**METHOD**

# SPIRAL: integrating and aligning spatially resolved transcriptomics data across different experiments, conditions, and technologies

Tiantian Guo[1,2†], Zhiyuan Yuan[3†], Yan Pan[4], Jiakang Wang[1], Fengling Chen[5], Michael Q. Zhang[6*] and Xiangyu Li[1*]

†Tiantian Guo and Zhiyuan Yuan contributed equally to this work.

*Correspondence:
michael.zhang@utdallas.edu;
lixiangyu@bjtu.edu.cn

[1] School of Software Engineering, Beijing Jiaotong University, Beijing 100044, China
[6] Department of Biological Sciences, Center for Systems Biology, The University of Texas, Richardson, TX 75080-3021, USA
Full list of author information is available at the end of the article

## Abstract

Properly integrating spatially resolved transcriptomics (SRT) generated from different batches into a unified gene-spatial coordinate system could enable the construction of a comprehensive spatial transcriptome atlas. Here, we propose SPIRAL, consisting of two consecutive modules: SPIRAL-integration, with graph domain adaptation-based data integration, and SPIRAL-alignment, with cluster-aware optimal transport-based coordination alignment. We verify SPIRAL with both synthetic and real SRT datasets. By encoding spatial correlations to gene expressions, SPIRAL-integration surpasses state-of-the-art methods in both batch effect removal and joint spatial domain identification. By aligning spots cluster-wise, SPIRAL-alignment achieves more accurate coordinate alignments than existing methods.

## Background

Recent years have witnessed the great success of single cell technologies in characterizing cell state dynamics in complex biological systems, which reveals the cellular mechanisms of development and disease at an unprecedented resolution. However, the lack of spatial dimension hindered the exploration of how cell identity and cell fate are influenced by their surrounding environment. Advances in spatially resolved transcriptomics (SRT) technologies have supplied the spatial dimension to achieve joint measurement of the gene expression profiles and spatial coordinates. Such spatial technologies include imaging-based [1–4] and next-generation-sequencing (NGS) -based technologies [5–8].

In the traditional single cell community, integrating single cell non-spatial transcriptomes from various sources enables the comparisons of distinct tissues, organs, or individuals in one shared space, e.g., the Human Cell Atlas (HCA) [9]. With the popularity of different spatial technologies in different biological systems and by different laboratories, the primary goal of HCA also extended to mapping those cells into common coordinate

maps (CCMs) [10] to analyze their functions and relationships in their spatial context. Like the shared low-dimensional spaces of integrated scRNA-seq data, in which each cell has its own position, a CCM is also a shared 2D/3D coordinate system, in which cells from distinct experiments can be embedded. Specifically, to create CCMs, we need to relate cells from distinct experiments with cells from a selected reference map, by which cells with distinct sources can be assigned new coordinates in the selected reference system. Besides, it is necessary to remove batch effect, caused by different samples, protocols, technologies, or laboratories, in SRT data with different origins in order to compare them in the shared feature space and in the shared coordinate space. Therefore, integrating SRT data includes two basic sequential tasks, i.e., removing batch effects and aligning coordinates. With the similarities of spots from different experiments supplied by the first task, one could accomplish the second task.

Unlike scRNA-seq technologies, in which cells' representations are classically defined by their own transcriptome profiles only, SRT technologies add spatial information, thus providing the opportunity to analyze the gene expression variations with both cell identity and the surrounding microenvironment [11]. However, the methods designed to remove batch effects of scRNA data [12–14] cannot be directly applied to SRT data. Firstly, these methods for scRNA-seq would eliminate the spatial dependences of the gene expressions that are needed not only in the compensation of low-quality or low-quantity sequencing of some experiments but also in the downstream analysis such as delineations of spatial domains and identifications of spatial genes. For example, the gene expression zonation gradients along the lobular axis of the liver revealed by spatial transcriptomes may be missed without spatial information [15]. Secondly, without spatial constraints, scRNA-seq based methods can easily mix up cells with similar gene expressions but distinct microenvironments. For example, the different sub-tumor microenvironments of pancreatic ductal adenocarcinoma have distinct influences on tumor immunity, subtypes, differentiation, and treatment response, which cannot be revealed only by transcriptome profiles alone [16].

Recently, several methods have been proposed to address the integration of SRT datasets containing multiple slices for joint clustering [17–20]. One such method, BASS [17], focuses on simultaneously performing cell type clustering and spatial domain detection for multiple samples using Bayesian hierarchical methods. However, BASS has two limitations. Firstly, it lacks low-dimensional embeddings or gene expressions without batch effects. Secondly, its performance in joint clustering for SRT data generated from different technologies is suboptimal. Another method, GraphST [18], tackles batch effect corrections for horizontally split slices using coordinate alignments through the PASTE algorithm [21] and for vertically split slices through manual coordinate alignment as the initial step. However, the effectiveness of GraphST heavily relies on the accuracy of coordinate alignments, which limits its widespread usage. DeepST [19] introduces a two-step process to remove batch effects. First, it utilizes a graph neural network for representation learning, which is followed by batch effect removal of these representations using domain adversarial neural networks (DAN). However, this approach cannot obtain the batch effect-free gene expression. Furthermore, the training methodology employed by DeepST hinders the effective integration of diverse samples. STAligner [20] achieves spatial-aware SRT data integration by combining STAGATE and a mutual nearest

neighbor (MNN)-based method into a unified model. Coordinate alignments are performed using spot pairs. The MNN-based method employed by STAligner aims to minimize distances between positive pairs and maximize distances between negative pairs. Consequently, it may overlook non-MNN pairs from the same spatial domains and mistakenly include MNN pairs belonging to different domains.

Besides these, some other methods [22, 23], focusing on spatial domain identifications, learn low-dimensional representations considering both gene expression and spatial information that can be used as inputs of Harmony [12] to obtain the integrated embeddings. However, the batch effect removal will be inadequate with separated processes of representation learning and batch effect removal, while the batch-effect-corrected gene expressions cannot be obtained either.

Besides the gene expressions, the spatial coordinates of different samples are also required to be aligned to make them comparable in the physical space. A few methods have tried to align coordinates of different SRT data recently. PASTE [21] attempts to align and integrate SRT data from multiple tissue slides using both gene expression and spatial information by computing pairwise similarities of adjacent slices with the fused Gromov-Wasserstein optimal transport. Through the pairwise alignments, PASTE achieves a stacked 3D structure of a tissue. It also creates the center slice from multiple similar slices. Nevertheless, PASTE has some limitations including the poor performance on non-replicate samples and the lack of batch-effect-corrected transcriptomes of all integrated samples. Alma et al. proposed another method to integrate different SRT data by constructing a common coordinate frame (CCF) to relate samples in physical space using Gaussian process regression (GPR) algorithm based on prior landmark knowledge [24]. The nonlinearity of GPR promotes its coordination alignment performance on samples with different cell compositions. However, the landmark cannot be always obtained in advance, and the batch-effect-corrected gene expression cannot be learned for downstream analysis.

To overcome the limitations of all these methods, we proposed a new method SPIRAL to effectively integrate data in both feature space, including low-dimensional embeddings, high-dimensional gene expressions, and physical space. SPIRAL consists of two seamless modules, i.e., SPIRAL-integration for batch effect removal and SPIRAL-alignment for spatial coordinate alignment. The first module (SPIRAL-integration) incorporates GraphSAGE [25] and domain adaptation [26] into a unified model to learn the corrected embeddings and expressions via combining transcriptome profiles and spatial contexts. The second module (SPIRAL-alignment) makes use of the results of SPIRAL-integration to align different coordinates via cluster-aware Gromov-Wasserstein distance [27]. GraphSAGE can encode features and spatial relationships into low-dimensional representations by inductively sampling and aggregating neighbors on large graphs. Domain adaptation can agglomerate different domains to a shared low-dimensional space. The combination of GraphSAGE and Domain adaptation achieves spatially aware integration of SRT data. The cluster-aware coordinate alignment can achieve accurate alignments of samples with distinct spatial structures.

We demonstrated the advantages of SPIRAL on several synthetic and recently published SRT datasets [8, 28–31]. On one hand, compared with conventional non-spatial methods, e.g., Seurat and harmony, SPIRAL achieves better or comparable integrations

in both the biological tissues containing the same set of spatial domains and the tissues containing distinct spatial domains. In contrast with Seurat and harmony, the learned embeddings of SPIRAL can reflect more locally continuous domains in multiple samples, and the batch-effect-corrected gene expressions present clearer spatial patterns and stronger functional domain enrichment. On the other hand, compared with the methods based on separated embedding and batch effect removal, e.g., harmony_SEDR and harmony_STAGATE, SPIRAL implemented more effective integrations in many experiments. More strikingly, SPIRAL outperformed the existing spatial based methods, DeepST, STAligner, GraphST, and BASS in our experiments. We further compared SPIRAL with coordinate alignments methods e.g., PASTE, in a diverse range of datasets, and found that SPIRAL output litter mixtures of different spatial domains, and clearer patterns of domain-specific genes. Importantly, SPIRAL can be generalized to unseen SRT datasets to predict their cluster labels and register them correctly in the reference coordinate systems (RCSs).

## Results

### Overview of method

In this work, we designed a method SPIRAL, which combines gene expressions and spatial relationships in the consecutive processes of batch effect removal and coordinate alignment by employing graph-based domain adaption and cluster-aware Gromov-Wasserstein optimal transport. The graph-based domain adaption is able to consider gene expressions of neighbors during removing batch effects, and the cluster-aware alignment can avoid mistakenly mixing up different clusters.

SPIRAL consists of two seamlessly connected modules, i.e., SPIRAL-integration and SPIRAL-alignment. SPIRAL-integration corrects batch effects via integrating expression profiles and spatial relationships with a unified framework combining GraphSAGE network [25] and domain adaptation network, and SPIRAL-alignment utilizes the clustering results of SPIRAL-integration to construct CCMs with cluster-aware Gromov-Wasserstein distance. SPIRAL-integration is composed of four neural networks (Fig. 1): (1) GraphSAGE functions as an encoder to encode gene expressions and spatial coordinates into a low-dimensional latent space; (2) noise classifier and (3) biology discriminator, they disentangle the low-dimensional embeddings to two parts: the noise part for distinguishing noise from different batches and the biology part for combining signals from different batches; (4) decoder network to reconstruct gene expressions. The biology part of the low-dimensional embedding is further grouped to uncover the spatial domains for multiple samples. Based on the clusters, SPIRAL-alignment assigns the spots or cells within the shared clusters of one sample to the positions of the corresponding ones of the reference sample. The reference sample is pre-selected to have the largest regions. The new coordinates of the sample-specific clusters in the reference coordinate system (RCS) are calculated via translation and/or rotation based on the spots of the shared clusters as datum points.

### SPIRAL achieves the effective integration on simulated data

To quantitatively evaluate the performance of SPIRAL on removing batch effects of the same cell types and discerning cells with similar expression profiling but distinct

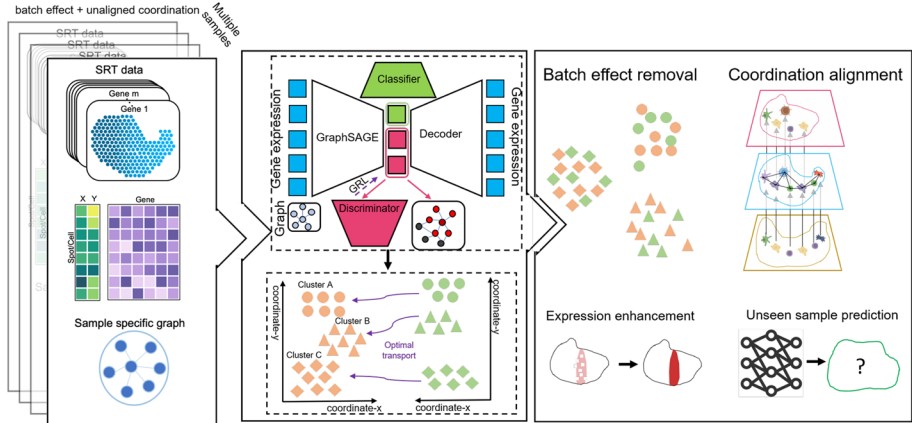

**Fig. 1** Overview of SPIRAL. SPIRAL includes two consecutive modules: SPIRAL-integration (upper right) and SPIRAL-alignment (bottom right). SPIRAL-integration takes combined gene expressions and separated graphs, constructed by spatial coordinates of spots for each sample, as input and learns the disentangled embeddings as biological embeddings and noise embeddings via a discriminator and a classifier. SPIRAL tries to preserve the spatial structures on biological embeddings and to preserve the expression patterns on the whole embeddings. SPIRAL-alignment takes cluster annotations of SPIRAL-integration and spatial coordinates as input to align coordinates of shared clusters between reference and query samples via Gromov-Wasserstein optimal transport. Through SPIRAL, the corrected embeddings, gene expressions, and aligned coordinates can be obtained. SPIRAL also can predict cluster labels and new coordinates for new SRT data

spatial neighbors, we generated three types of simulated datasets (simulate1-3) with the same cell-type compositions but different spatial arrangements, in which two cell types (Group1 and Group2) are adjacent in feature space and distant in physical space (Additional file 1: Fig. S1A). We used Splatter [28] to simulate gene expressions and assigned coordinates to each cell in the restricted regions (Methods). We evaluated the efficiency of batch effect removal by the combined local inverse Simpson index (LISI-CoM) [12] measuring the ability of both mixing same clusters and separating different clusters (Methods). We also employed adjusted Rand index (ARI) [32] to assess the correspondences of the annotated groups and the learned clusters with the batch-effect-corrected embeddings. The Uniform Manifold Approximation and Projection (UMAP) visualizations (Fig. 2A and Additional file 1: Fig. S1B) and the quantitative measures LISI-CoM and ARI (Fig. 2B) proved the superior performances of SPIRAL-integration on the three datasets when compared to the other seven methods: harmony [12], Seurat [13], harmony_STAGATE [22], harmony_SEDR [23], GraphST [18], DeepST [19], STAligner [20]. It is noted that, for BASS, the lack of integrated embedding and gene expression prevented the calculation of LISI-COM and iLISI. The ARI of SPIRAL-integration were higher than the multi-sample clustering method BASS [17] on all datasets (Fig. 2B right). As expected, only the spatial-based methods SPIRAL, harmony_STAGATE, and STAligner were able to discern Group1 and Group2, while harmony_STAGATE and STAligner were not able to remove batch effects (Fig. 2A). Then, we utilized the clusters of SPIRAL-integration to assign the cells from Batch2 to the spatial coordinates of Batch1, which is selected as the RCS (Additional file 1: Fig. S1A). We evaluated the spatial coherence using the spatial coherence score (SCS) of PASTE [21], which measures the probability of the spots in the neighborhood having the same clusters compared with the random assignment of the clusters. SPIRAL-alignment achieved higher SCS than PASTE

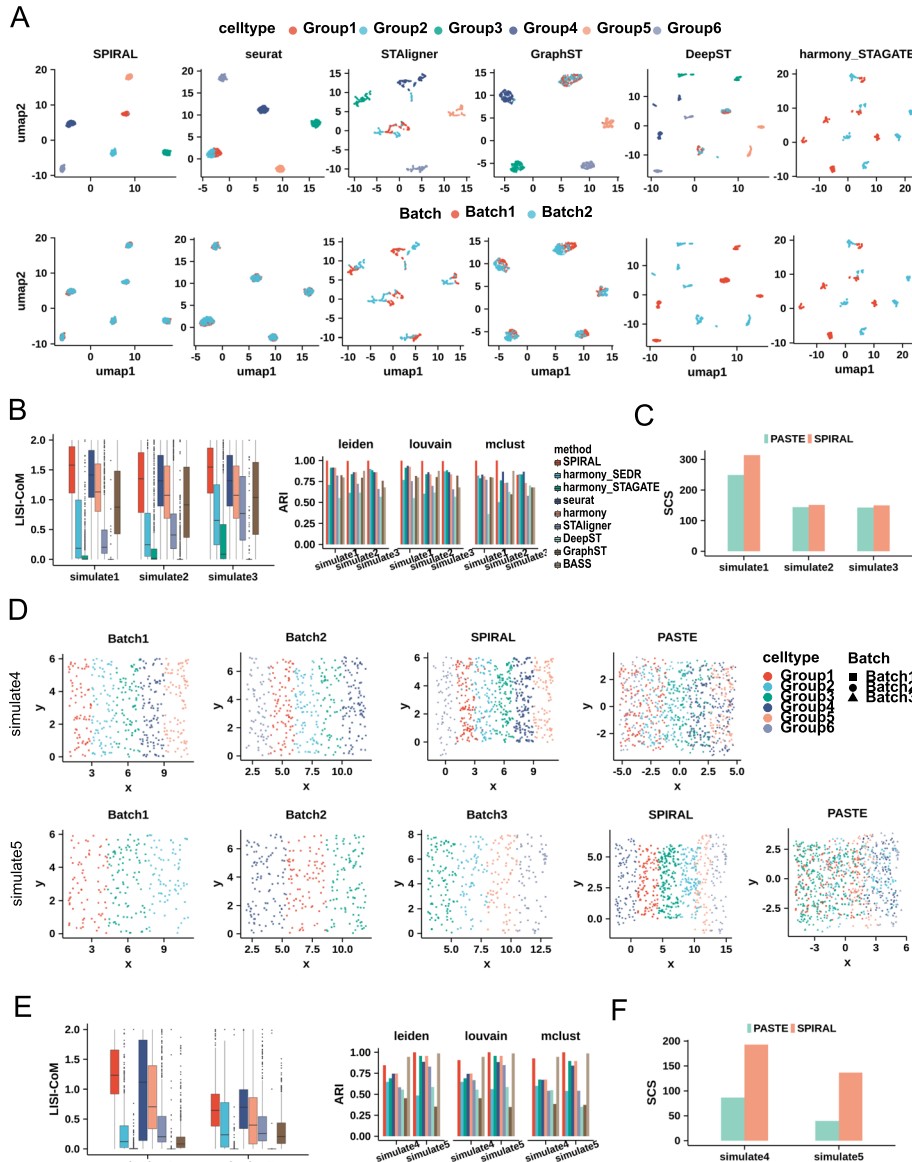

**Fig. 2** Simulating. **A** UMAP visualizations of simulate1 data. Colors represents cell types (upper) and batches (bottom), and each column responds to each method. **B** Boxplot of batch-effect-removing accuracy (LISI-CoM) (left) and clustering accuracy (ARI, Louvain clustering) (right) in three datasets: simulate1, simulate2, and simulate3 for eight or nine methods. **C** The spatial coherence score (SCS) of annotated cell types of SPIRAL and PASTE in three datasets. **D** spatial arrangements of cell types on original and aligned coordinates by SPIRAL and PASTE in simulate4 (upper) and simulate5 datasets (bottom). **E** The quantitative measurements of LISI-CoM and ARI in simulate4 and simulate5 datasets for eight or nine methods. **F** SCS of cell types in simulate4 and simulate5 data for SPIRAL and PASTE

in all three datasets (Fig. 2C). It is worth noting that SPIRAL-alignment could overcome the nonlinear transformations between two bathes, as the dataset simulate2 had different coordinate rotations between the batches and simulate3 had coordinate scaling on the basis of simulate2 (Additional file 1: Fig. S1A).

We also compared the methods in more complex scenarios, such as distinct cell compositions. We simulated a two-batch dataset (simulate4) containing Group1-4 as

batch-shared cell types and Group5-6 as batch-specific cell types, and another three-batch dataset (simulate5) containing Group1-3 in Batch1, Group1,3,4 in Batch2, and Group2,3,5,6 in Batch3 (Fig. 2D). We assumed there were the same spatial arrangements among the shared cell types. SPIRAL achieved the best performances on both datasets considering all the measurements (Fig. 2D–F and Additional file 1: Fig. S1C). Specifically, in terms of coordinate alignments, PASTE had poor performances by mixing the cells with different cluster labels, which indicates PASTE's limitations in aligning samples with different cell compositions.

## SPIRAL achieves consistent integrations and coordinate alignments across donors of the human dorsolateral prefrontal cortex data (DLPFC)

We then applied SPIRAL on a 10 × Genomics Visium dataset DLPFC, which contains 12 slices from three independent donors [29] (Fig. 3A). Each donor has two pairs of "spatial replicates," in which each pair is composed of 10-um adjacent slices, and the second pair is located 300 μm posterior to the first pair. We referred to the four slices as "A, B, C, D" sequentially, in which AB and CD are the two replicates and BC are 300 μm apart (Fig. 3A). Maynard et al. [29] have provided the spot-wise annotations ranging from white matter (WM) to different cortical layers in this dataset, which were used as the ground truth in the following evaluations. It is worth noting that all the 12 slices came from DLPFC of the neurotypical adult donors, whose differences were mainly caused by batch effects.

In total, we conducted 13 experiments to integrate each pair of consecutive slices, such as "AB," "BC," and "CD," and four slices 'ABCD' within each donor, and did experiment to integrate all slices across donors to compare methods in terms of similar or slightly different samples. The quantitative measurement LISI-CoM demonstrated the superiority of SPIRAL in correctly mixing up the same functional layers while separating the different ones (Fig. 3B left). Moreover, SPIRAL demonstrated superior performance in joint clustering on the integrated samples from the 13 experiments. It achieved the highest adjusted Rand index (ARI) regardless of the employed clustering methods, including Louvain [33], Leiden [34], and mclust methods [35] (Fig. 3B right). When comparing with the other methods, we found that harmony_STAGATE had the lowest LISI-CoM

(See figure on next page.)

**Fig. 3** DLPFC. **A** Schematic of DLPFC data, which included four sequential sections A, B, C, and D of three donors. **B** The accuracy of batch effect removing and Louvain clustering, LISI-CoM (left) and ARI (right), in 13 integration experiments. **C** UMAP visualizations of 12-section integrations by different methods. Colors represent cell types (upper) and batches (bottom). **D** The barplot of Pearson correlation between deduced pseudo-time and annotated layers for 12-section integration experiment. **E** The boxplot of SCS of annotated layers on aligned coordinates by SPIRAL and PASTE in the 12 experiments. The "ns" means not significant differences. **F** Layers assignments on aligned coordinates by SPIRAL and PASTE for four sections of each donor. Colors represent different layers (upper) and shapes represent different sections. **G** SCS of annotated layers on aligned coordinates by SPIRAL and PASTE for 12-section integrations. **H** Spatial autocorrelations (Geary's *C* and Moran's *I*) of raw, SPIRAL, and Seurat-integrated gene expressions on original coordinates and raw and SPIRAL-integrated expressions on aligned coordinates, labeled as Raw, SPIRAL and Seurat, Raw-AC and SPIRAL-AC. **I** Violin plot of TRABD2A in annotated layers (upper) and deduced clusters (mclust, bottom). **J** spatial arrangements of raw (upper), Seurat (middle), and SPIRAL (bottom)-integrated gene expressions in original three spatial coordinates (the first three columns from the left) and aligned coordinates (the fourth column from the left)

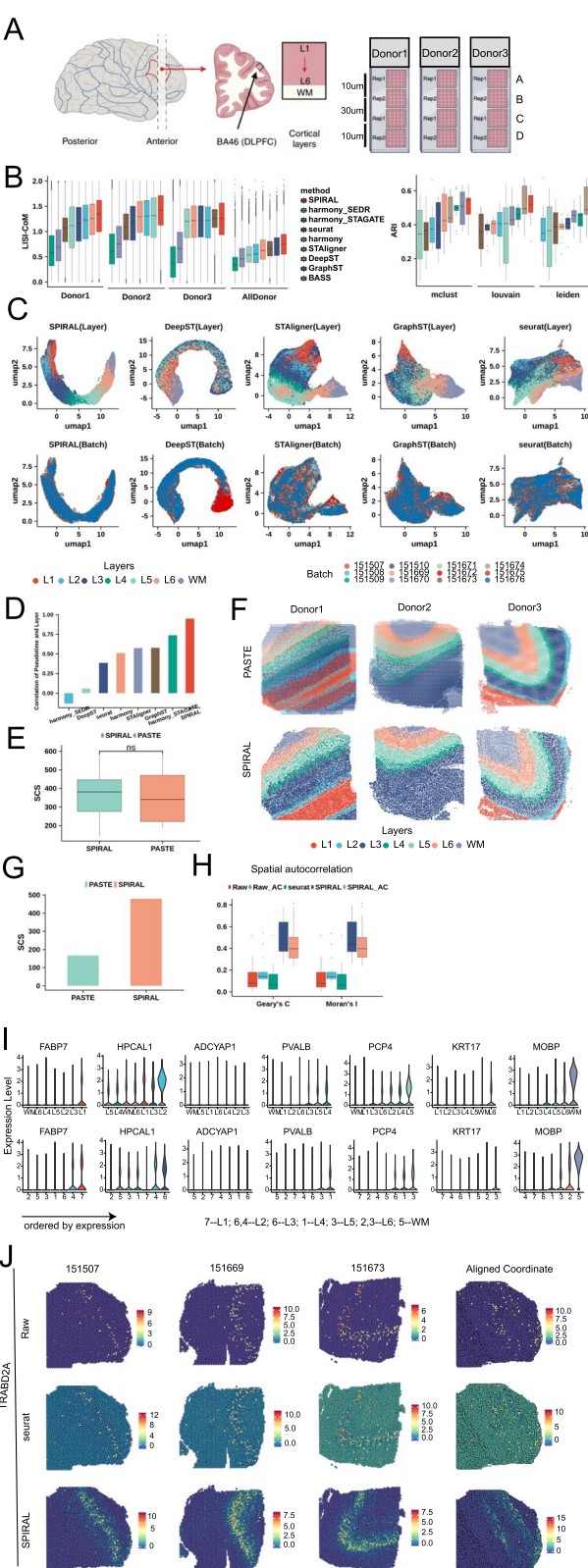

**Fig. 3** (See legend on previous page.)

in all examinations, though it had higher ARI in many experiments. In contrast, STAligner achieved higher LISI-CoM and similar joint clustering accuracy compared to harmony_STAGATE through inserting batch-effect-removal process into the unified model (Fig. 3B). We displayed the UMAP visualizations of five methods in Fig. 3C and Additional file 1: Fig. S2A; we found that, as a whole, all these methods realized effective integrations of different donors, while except for SPIRAL, the remaining methods had some wrong mixtures of the different layers. From the layouts of these UMAP embeddings, we found SPIRAL was the only method to accurately delineate a clear and correct spatial trajectory from layer1 to WM. To measure to what extent the low-dimensional embeddings could depict the trajectory from layer1 to WM quantitatively, we quantized the layers from layer1 to WM as 1 to 7, whose Pearson correlations with PAGA derived pseudo-time were calculated [36] (Methods). We displayed the Pearson correlations of eight methods in Fig. 3D and found substantial improvements of SPIRAL in spatial trajectory preservation, compared with the other methods.

Furthermore, we proceeded to integrate the four slices from each donor to establish their common coordinate systems (Fig. 3E and F). In comparison to PASTE, SPIRAL demonstrated comparable spatial coherence of the layers in the majority of the experiments and exhibited higher accuracy in aligning the four slices of donor1, which were more distinct from each other (Fig. 3F). Additionally, we tried to align coordinates of 12 slices across the three donors. To align the spatial coordinates of all the samples, we selected sample 151,507 as the reference and mapped all other samples to the reference coordinate system. The spatial coherence of the annotated layers with the aligned coordinates of SPIRAL-alignment was much higher than that of PASTE, indicating the superiority of SPIRAL-alignment in the alignment of different individuals (Fig. 3G).

We further utilized the batch-effect-corrected gene expressions (Methods) to verify the effectiveness of SPIRAL in downstream analysis. Firstly, we examined the spatial expression patterns of the layer-marker genes (proposed in the original paper [29]), in terms of raw and integrated expressions. We used Moran's $I$ and Geary's $C$ to measure the spatial autocorrelations of the gene expressions [37, 38] and found that SPIRAL produced much clearer spatial patterns than the raw and Seurat-integrated expressions in the original physical spaces (Fig. 3H). We next used the combinations of SPIRAL-integrated expressions or raw expressions and SPIRAL-aligned coordinates to calculate Moran's $I$ and Geary's $C$. The much better spatial coherence of the new aligned coordinates than on the original coordinates (Fig. 3H) indicates that the common coordinate system could better recover the underlying spatial patterns consistent with the layer-marker genes.

We then further demonstrated the preservation of the laminar organization on the new coordinates in terms of the spatial distributions of both the annotated layers and the layer-marker gene expressions. We displayed the annotated layers on the aligned coordinates of SPIRAL and PASTE (Additional file 1: Fig. S2B), where multiple false mixtures of different layers were found in PASTE and relatively much fewer false mixtures among L1-3 or L4-6 in SPIRAL, which was almost inevitable in the continuous laminar organizations. Additionally, we showed the spatial dispositions of our clusters, derived by the clustering algorithm mclust [35], in Additional file 1: Fig. S2B, which delineated the true laminar structures of the human cortex. To annotate these clusters, we compared the expressions of the layer-marker genes of the annotated layers with our clusters (Fig. 3I),

and we found that almost all of them could be matched with the true layers (7 vs L1, 4 vs L2, 6 vs L3, 1 vs L4, 3 vs L5, 2 vs L6 and 5 vs WM). Finally, we compared the enrichment of the layer-marker genes expressed in raw data, SPIRAL-integrated data, and Seurat-integrated data (Fig. 3J and Additional file 1: Fig. S2C); we found that SPIRAL could denoise and impute genes as its integrated expressions were more clearly and exclusively enriched in certain layers.

### SPIRAL realizes a complete depiction of the whole sagittal mouse brain

Next, we tested whether SPIRAL could analyze tissue sections with more complicated structures (Fig. 4A). We applied SPIRAL to sagittal mouse brain data generated by 10X Visium protocol, which includes one pair of replicates in both anterior and posterior brain separately, labeled as "anterior1/2" and "posterior1/2" (Additional file 1: Fig. S3A). We conducted three integration experiments: integrating anterior1 and anterior2, integrating posterior1, and posterior2 and integrating both anterior1/2 and posterior1/2. Given the absence of confidential annotations serving as ground truth, we utilized iLISI [12] to evaluate the extent of sample mixtures. To avoid the false mixtures of different domains be calculated by iLISI in integrating anterior1/2 and posterior1/2 sections, the iLISI was calculated using spots located on the 250- or 300-pixel distances around adjacent regions. According to the iLISI evaluations (Fig. 4B left) and UMAP visualizations (Additional file 1: Fig. S3D), SPIRAL-integration and Seurat emerged as the top two methods in terms of integration performance across all three experiments.

We further measured the consistency of the Louvain clusters obtained by nine methods with the spatial arrangements via SCS [21]; the bar plot showed that SPIRAL-integration and harmony_STAGATE had comparable values, which were slightly smaller than STAligner and much larger than the remaining methods on two integration experiments (Fig. 4B right). Combining iLISI and SCS, we concluded that SPIRAL-integration performed best on both batch effect correction and spatial dependency preservation. Next, we aligned the coordinates of anteiror1 and anterior2, posterior1, and posterior2, with coordinates of anterior2 or posterior2 were rotated 30, 60, 90, 120, 150, and 180 degrees (60-degree rotation was shown in Additional file 1: Fig. S3E), using SPIRAL-alignment and PASTE separately. To measure the spatial coherence of domains on the aligned coordinates, we annotated the domains (Additional file 1: Fig. S3B) using the domain-marker genes (Additional file 2) and hematoxylin and eosin (H&E) stained

(See figure on next page.)

**Fig. 4** Sagittal mouse brain. **A** The anatomy diagram of anterior and posterior mouse brains to outline the task: integrating both mouse anterior and mouse posterior to delineate the whole sagittal mouse brain. **B** Quantitative measurement of iLISI for eight methods (left) and SCS of derived clusters for nine methods (right), in three integration experiments: anterior1 and anterior2, posterior1 and posterior1, both anterior1/2, and posterior1/2. **C** Distribution of SCS of annotated domains on aligned coordinates by SPIRAL and PASTE for six-category rotated anterior replicates and posterior replicates alignments. **D** Spatial arrangements of Louvain clusters derived by embeddings of four methods for four samples. **E** Spatial distributions of raw (upper), Seurat (middle), and SPIRAL (bottom) integrated gene expressions in cortex region for anterior1 and posterior1/2. **F** Spatial autocorrelation of raw, Seurat, and SPIRAL-integrated domain-marker genes in four sagittal mouse brain samples. **G** Wilcoxon rank sum test (−10log(pvalue)) of fold changes of domain-marker gene expressions in relative domains compared with domains having the second highest expressions

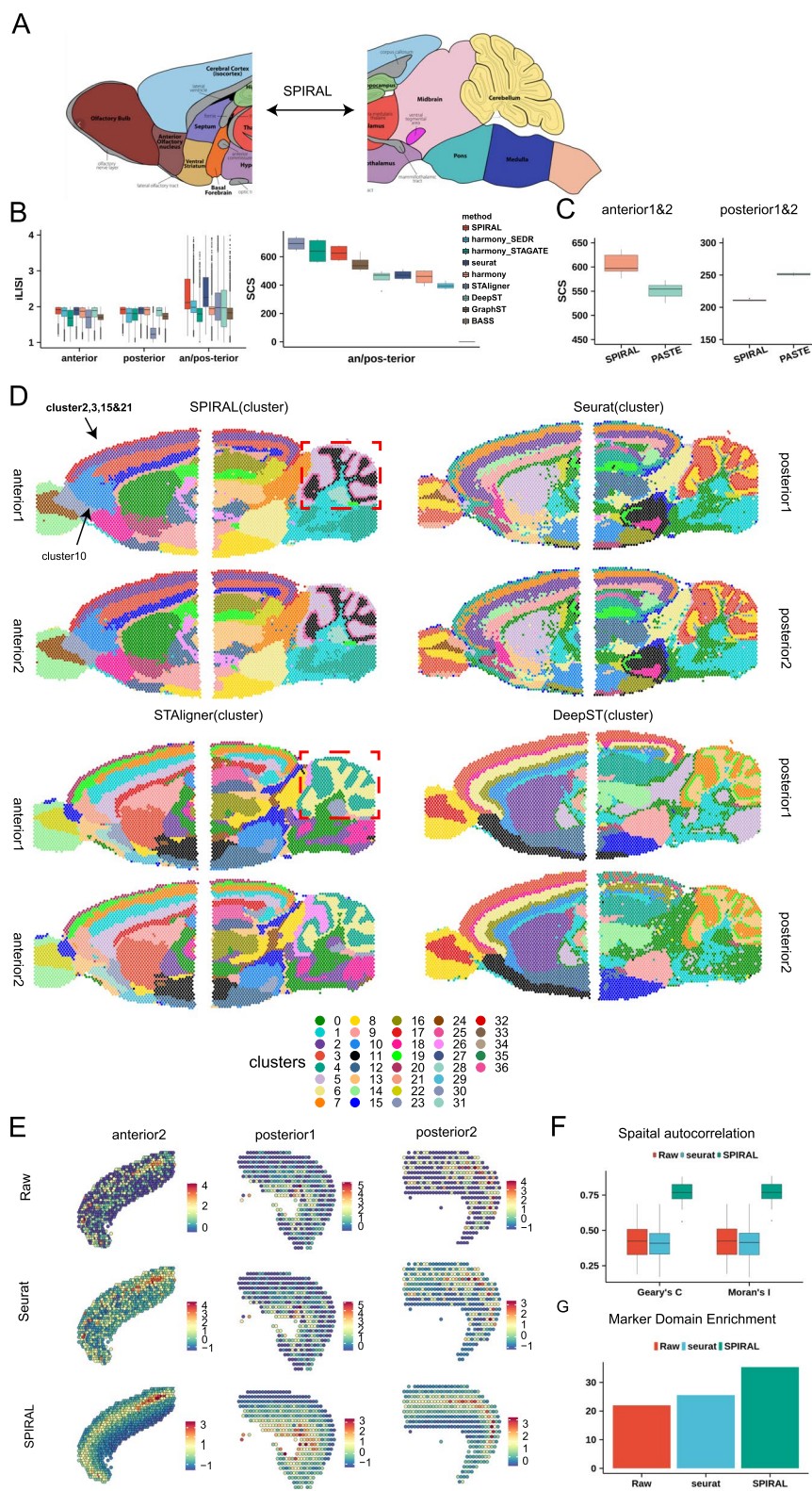

**Fig. 4** (See legend on previous page.)

images (Additional file 1: Fig. S3A and C). The SCS of the annotated domains of both experiments (anterior: median 597 vs 554; posterior: median 210 vs 251) proved SPIRAL-alignment and PASTE had close performances in aligning the replicates (Fig. 4C and Additional file 1: Fig. S3F).

We then displayed the Louvain clustering results derived by embeddings of four methods in Fig. 4D respectively, where SPIRAL, STAligner, and DeepST had more continuous boundaries between domains and fewer outliers than non-spatial methods Seurat. Moreover, SPIRAL, STAligner, and Seurat had almost the same spatial distributions of clusters between replicates anterior1 and anterior2 or posterior1 and posterior2 and identified the shared clusters among the adjacent regions of the two consecutive sections, anterior1/2 and posterior1/2, which mainly included cortex, hippocampus, thalamus, and hypothalamus regions (Fig. 4D), whereas DeepST failed to identify common domains in the cortex region between posteiror1 and posterior2, as well as between anterior2 and posterior2 (Fig. 4D). Additionally, compared to STAligner, even with fewer clusters (STAligner: 31 vs SPIRAL: 29), SPIRAL was able to identify finer structures, such as GL_interneuron and PL_interneuron of cerebellum (red box of Fig. 4D).

We further analyzed the clusters of SPIRAL and found some sub-clusters of Louvain method compared with mclust method (Additional file 1: Fig. S3G). For example, in the anterior brains, cluster10 was separated from the cortex layers (cluster2, cluster3, cluster15, cluster21) and was not included in the posterior brains. The differential expressed genes (DEG) of cluster10 were more enriched in the olfactory region (Additional file 1: Fig. S3H, Wilcoxon rank sum test: $P$-value $< 2.2e-16$) than the other clusters (cluster2, cluster3, cluster15, cluster21) of the cortex. Combining the DEGs and the mouse brain sagittal anatomy diagram (Additional file 1: Fig. S3B), we deduced that this sub-cluster belonged to the anterior olfactory cortex (AOC). This sub-cluster could not be revealed by Seurat, which may further indicate the importance of spatial information during correct and functional integration.

Additionally, we testified the advantages of SPIRAL in terms of the batch-effect-correction gene expression profiles. Firstly, SPIRAL was able to denoise and impute gene expressions, which could be proved by the clear expressions of domain-marker genes, such as Rorb, a marker of the cortex layer4, in anterior2 and posterior2, and the layer6-marker Hs3st4 in posterior1 (Fig. 4E). Secondly, SPIRAL could generate gene expressions displaying more coherent spatial patterns, which were measured by higher Moran's $I$ and Geary's $C$ of domain-marker genes (Additional file 2) than raw and Seurat-integrated data (Fig. 4F). Finally, the integrated expressions of domain-marker genes had higher expression enrichment in relative domains than in the other domains (Fig. 4G, Methods).

**SPIRAL can integrate SRT samples with different experimental protocols**

In the previous sections, we demonstrated SPIRAL's integration performance on data generated by similar experimental protocols. In this section, we applied SPIRAL to datasets generated by more different protocols (Fig. 5A). These datasets included coronal mouse brains sequenced by 10X Visium technology with different preserving and staining methods: formalin-fixed paraffin-embedded (FFPE) tissue with H&E staining, fresh frozen tissue with H&E staining, and fresh frozen tissue with immunofluorescence

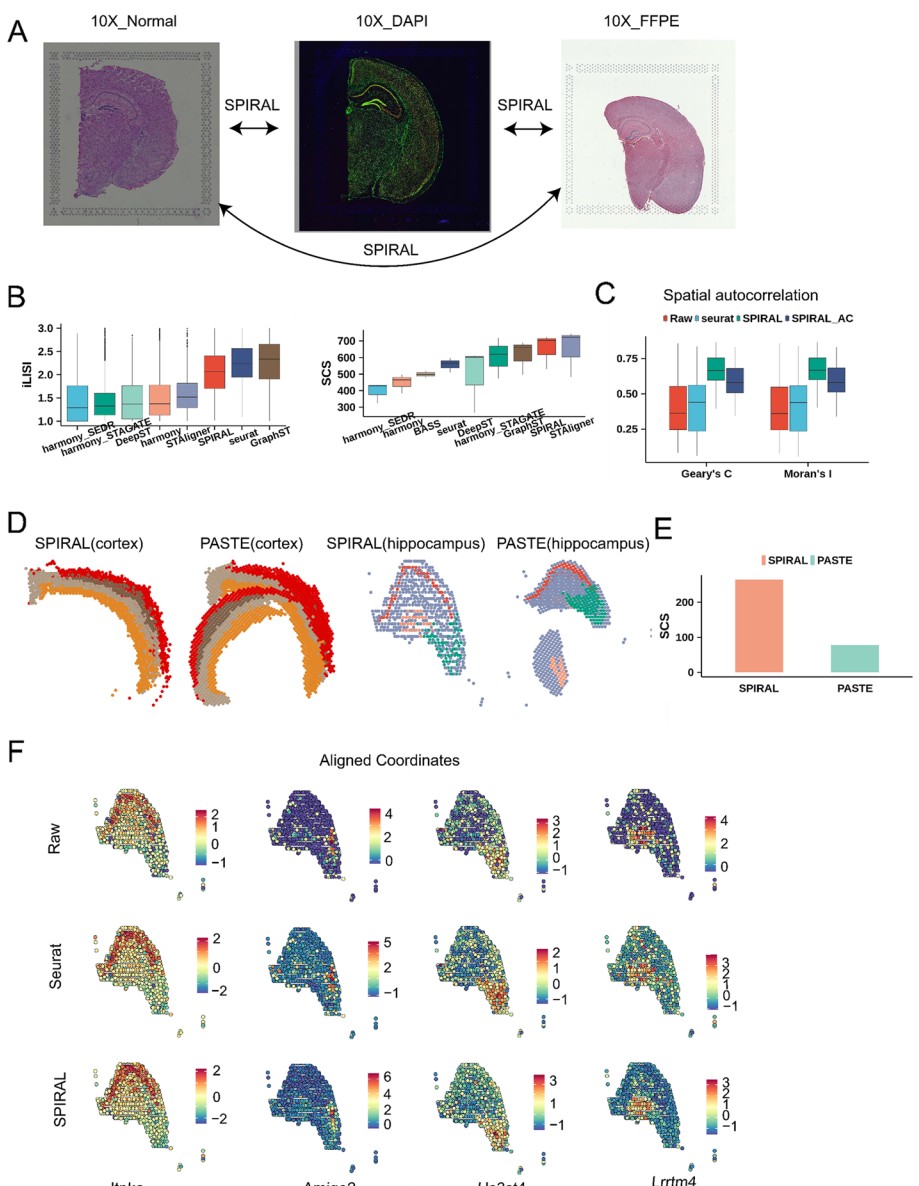

**Fig. 5** Coronal mouse brain. **A** H&E staining images of 10X_Normal and 10X_FFPE data and DAPI staining image of 10X_DAPI data were displayed to illustrate this experiment: integrating three datasets with different protocols. **B** Boxplot of iLISI (left) and SCS of derived clusters (right) in three-coronal-mouse-brain integration experiment. **C** Spatial autocorrelations of raw, Seurat and SPIRAL-integrated expressions on original or aligned coordinates for coronal mouse brain dataset. **D** Spatial arrangements of annotated domains on aligned coordinates of SPIRAL and PASTE in cortex and hippocampus regions. **E** SCS of annotated layers on aligned coordinates by SPIRAL and PASTE in coronal mouse brain dataset. **F** Domain-marker gene expression profiles on aligned coordinates by SPIRAL in hippocampus of coronal mouse brain

(IF) staining, labeled as "10X_FFPE," "10X_Normal," and "10X_DAPI." We annotated domains using marker genes in each sample (Additional file 2), where the cortex, hippocampus, and thalamus were identified (Additional file 1: Fig. S4A).

From the quantitative measurements of iLISI, SCS and UMAP visualizations (Fig. 5B and Additional file 1: Fig. S4B), we concluded that SPIRAL and Seurat could remove

batch effects effectively, while other methods were unable to integrate FFPE sample with the two frozen fresh samples. We noted that Seurat had higher iLISI (mean: 2.23 vs 2.05 in Fig. 5B left) and lower SCS (mean: 557 vs 653 in Fig. 5B right), Moran's *I* (mean: 0.424 vs 0.666) and Geary's *C* (mean: 0.424 vs 0.665) than SPIRAL (Fig. 5C). These results demonstrated clusters identified by methods missing spatial relationships could not represent spatial domains, which were defined as having high spatial coherence in both gene expression and histology [39, 40]. Furthermore, we specifically focused on the hippocampus region, where the "cord-like" and "arrow-like" structures serve as visible landmarks outlined by the Allen Brain Atlas [41] and domain-marker gene expressions (Additional file 1: Fig. S4C). By examining the cluster arrangements derived from the embeddings of four spatial-based methods in the hippocampus region, we observed that SPIRAL outperformed other methods in delineating the dentate gyrus (DG) and the CA1/2 and CA3 layers of the hippocampus. These corresponded to cluster2, cluster1, and cluster3, respectively, across all three samples (Additional file 1: Fig. S4D). In contrast, harmony_STAGATE and STAligner exhibited poorer performances on the '10X_FFPE' sample by missing the CA1/2 layers. Notably, DeepST failed to capture the 'arrow-like' structure on all samples, particularly on the '10X_FFPE' sample (Additional file 1: Fig. S4D).

We further aligned spatial coordinates across three samples, where we selected fresh frozen tissue with H&E staining ("10X_Normal") as the reference coordinate map on account of its intact structure and high-quality gene expressions (Additional file 1: Fig. S4A and E). SPIRAL showed more accurate coordination alignments than PASTE, both in aligning the main functional domains, i.e., cortex (Fig. 5D, left) and hippocampus (Fig. 5D, right), and in aligning the whole samples (Additional file 1: Fig. S4F). The quantitative result of SCS also showed SPIRAL's better performance than PASTE (Fig. 5E).

We then visualized the spatial distributions of gene expressions of the "cord-like" and the "arrow-like" structures within the hippocampus (Fig. 5F) and of the cortex layers (Additional file 1: Fig. S4G) in the aligned coordinates. We observed that the domain-marker genes displayed enrichments in the corresponding domains whether of raw expression or Seurat- or SPIRAL-integrated expression, and SPIRAL-integrated genes had clearer spatial patterns. These results demonstrated the spatial patterns of gene expressions were preserved on the aligned coordinate system of SPIRAL, suggesting the feasibility to register new coronal mouse brain SRT data in the aligned coordinate map. Additionally, the integrated gene expressions of SPIRAL on aligned coordinates displayed clearer spatial patterns by the higher Moran's *I* and Geary's *C* than Seurat and raw (Fig. 5C).

**SPIRAL can integrate SRT samples with different spatial technologies**

Finally, we testified whether SPIRAL could integrate datasets with different spatial resolutions using SRT data of mouse olfactory bulb (mouse OB) sequenced by 10X Visium, Stereo-seq [30], and Slide-seq V2 [31], whose resolutions were 50 μm, 35 μm (bin50, $50 \times 50$ DNA nanoball), and 10 μm, labeled as ("10X Visium," "Stereo-seq," and "SlideV2") respectively (Fig. 6A). In this experiment, we integrated data of Stereo-seq and Slide V2 as a reference to register 10X Visium data (Fig. 6A). We showed the iLISI of the five methods in Fig. 6B and found SPIRAL-integration achieved the best performances in

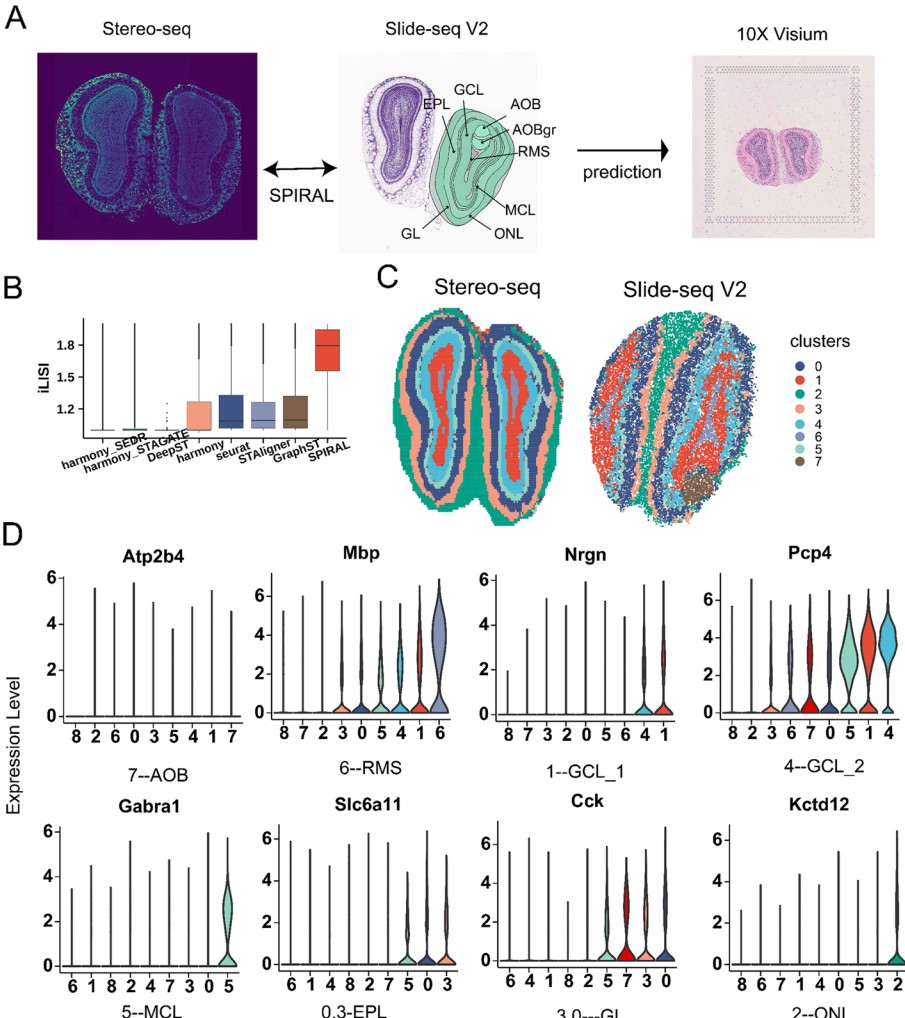

**Fig. 6** Mouse olfactory bulb. **A** laminar structures of Allen reference atlas with functional layers were annotated and H&E staining of 10X Visium and Stereo-seq data, this experiment aimed to integrate Stereo-seq data and Slide-seq V2 data, whose model were used to predict 10X Visium data. **B** Boxplot of iLISI-CoM of eight methods in integrating mouse olfactory bulb datasets of Stereo-seq and Slide V2-seq. **C** Joint clustering (Louvain) for integrated Stereo-seq and Slide V2-seq datasets by SPIRAL. **D** Violin plot of domain-marker gene expressions in derived clusters by SPIRAL

batch integrations. The spatial arrangements of spatial clusters by SPIRAL-integration delineated the structures of mouse OB, whose layer-marker genes and spatial positions suggested the correspondences between typical structures of mouse OB and Louvain clusters: "7" vs accessory olfactory bulb (AOB), "6" vs rostral migratory stream (RMS), "1" vs granular cell layer (GCL_1), "4" vs GCL_2, "5" vs mitral cell layer (MCL), "0" vs external plexiform layer (EPL), "3" vs glomerular layer (GL), and "2" vs olfactory nerve layer (ONL) (comparing Fig. 6A, C, D). The domains identified by harmony_STAGATE revealed its inability to perform joint domain identifications, resulting in the failure to detect finer structures like the RMS in the low-resolution Stereo-seq data. This limitation persisted even when the Louvain resolution was increased from 1.0 to 1.5 (Additional file 1: Fig. S5B). This observation was alleviated by STAligner (Additional file 1:

Fig. S5C), which identified some common clusters, such as 8, 3, 0, 1, and 2, corresponding to RMS, GCL_1, GCL_2, MCL, and GL. However, other common structures, such as ONL and EPL, were not accurately identified in a joint manner. The results of spatial alignment of Stereo-seq and Slide-seq V2 data with divergent coordinate scales demonstrated the superiority of SPIRAL (Additional file 1: Fig. S5A).

After integrating Stereo-seq and Slide-seq V2, we wanted to test the abilities of prediction and generalization of our model. To this end, we used the trained model to predict the 10X Visium data. The spatial distributions of the predicted clusters were well consistent with the anatomic structures (Additional file 1: Fig. S5D and Fig. 6A). We further registered the spots of 10X Visium data to the reference coordinates of Stereo-seq data, whose spatial arrangements of predicted clusters corresponded well with typical laminar organizations from GCL in the inner layer to ONL in the outer layer (Additional file 1: Fig. S5E). To assign the neurons of mouse olfactory bulb to domains, we predicted clusters of scRNA data [42] using a model trained by Stereo-seq and Slide-seq. We constructed a graph of scRNA by the similarities of the low-dimensional gene expressions. The percentages of cell types in each layer corresponded well with the prior knowledge, for example, the granule cells are in GCL, mitral and tufted cells are in MCL, external plexiform layer neurons in EPL, periglomerular cells in GL, and olfactory sensory cells in ONL [43–45] (Additional file 1: Fig. S5F).

We finally testified the spatial autocorrelations of the raw and integrated gene expressions of SPIRAL and Seurat using Moran's $I$ and Geary's $C$ and found that our integrated genes reflected the spatial patterns more clearly, whether in the original coordinate maps or in the aligned coordinate maps (Additional file 1: Fig. S5G). The difference of the integrated and the raw domain-marker gene expression profiles further indicated the ability of SPIRAL to denoise and impute gene expressions (Additional file 1: Fig. S5H).

## Discussion

With the increasing development of spatial transcriptome technologies, more and more SRT data will be generated, which brings a great need for integrating and aligning these SRT data. Here, we proposed a method SPIRAL, to consecutively perform spatially batch effect removal and nonlinear coordinate alignment by integrating gene expression profiles and spatial contexts. SPIRAL is composed of two connected modules: SPIRAL-integration for batch effect removal and SPIRAL-alignment for spatial coordinate alignment. By combining inductive graph neural network GraphSAGE and domain adaptation network, SPIRAL-integration could preserve spatial structures and eliminate unexpected discrepancies across samples, which also could be generalized to unseen SRT data. By utilizing the inferred clusters by embeddings of SPIRAL-integration, SPIRAL-alignment could accomplish cluster-wise coordinate alignments, which were more accurate than linear methods.

In this study, we applied SPIRAL to both simulated and several published datasets, which includes data from different experiments, different protocols, and different SRT technologies. All these data demonstrate the superior of SPIRAL in removing batch effects and aligning coordinates, especially for samples with considerable differences in cell compositions and spatial structures. SPIRAL also denoised and imputed gene expressions to have clearer spatial patterns and stronger domain enrichments by

incorporating the spatial context of multiple samples. These genes were also proved to have biological functions. Finally, SPIRAL could be applied to unseen SRT data to predict their cluster labels and their new coordinates in reference coordinate systems.

However, SPIRAL can still be improved in the following aspects. Firstly, SPIRAL could further combine histological image, spatial locations, and gene expressions to remove batch effects both in gene expressions and in images, aligning both spots and pixels. This would enhance the delineation of morphological structures across multiple samples [46]. Secondly, SPIRL-alignment could be improved to ensure a more evenly distributed alignment within each cluster. Thirdly, while SPIRAL performs well with current datasets, its utility could be extended to higher resolution data, such as MERFISH [4] data, to achieve more precise delineation of structures. Additionally, parallel computation and distributed learning could be utilized to accelerate training procedures [47, 48]. Moreover, as spatial multiomics data become increasingly available, integration strategies across various spatial omics would be desperately needed. SPIRAL could expand its capabilities to integrate a variety of spatial multimodal data, such as FISH-based SRT data and spatial proteomics data [49, 50].

## Conclusions

In this contribution, we have presented SPIARL for multi-sample batch effect removal, domain identification, and spatial coordinate alignment in SRT data across different experiments, conditions, and technologies. In comparison with the existing methods, SPIRAL-integration achieves more thorough data integration and spatial structure preservation, SPIRAL-alignment produces more accurate coordinate alignment. In conclusion, as an effective and convenient tool, SPIRAL facilitated multi-sample joint analysis of SRT data.

## Methods

### SPIRAL algorithm description

#### GraphSAGE

GraphSAGE (graph sampling and aggregating) is a deep neural network-based inductive graph node embedding framework, whose inductive learning mode superiorities general graph convolution network (GCN)-based methods on predictions of unseen data, which is the HCA intends to do for more and more unseen scRNA and SRT data using current data. GraphSAGE leverages node features and graph structures to learn an embedding function to embed nodes to low-dimensional spaces. In our framework, GraphSAGE functions as an encoder to generate low-dimensional embeddings by nonlinearly aggregating gene expressions from its physical neighbors. We first convert SRT data into an undirected graph, $G^d\left(V^d, E^d\right)$ for $\{d = 1, \ldots, D\}$ samples. In graph $G^d\left(V^d, E^d\right)$, each node represents a spot, labeled as $v_{id} \in V^d$ for spot $i$, and the edge $e_{ijd} \in E$ represents the spatial correlation of spots $i$ and $j$ in sample $d$. We proposed two alternative methods to construct graph, knn-based and radius-based methods. We used knn-based method in this paper, and we set knn = 6 for 10X Visium data, knn = 8 for Slide-seq V2 and Stereo-seq data, and knn = 10 for synthetic data. Let $x_{id} \in R^M$ be the normalized and minmax scaled gene expression of spot $i$ in sample $d$, where $M$ represents the number of selected genes and $X = \{x_{id}, \forall i = 1, \ldots, S_d, d = 1, \ldots, D\}$. We set $K$ as the

number of layers of GraphSAGE. GraphSAGE takes $X$ and $G(V,E) = \{G^d(V^d, E^d), \forall d = 1, \ldots, D\}$ as input, and let $X$ be the initial embedding (i.e., $h_{id}^0 = x_{id}, \forall i = 1, \ldots, S_d, d = 1, \ldots, D$). For spot $i$ of graph $d$, its $k$th($k = 1, .., K-1$) embedding $h_{id}^k$ is obtained by aggregating embeddings $h_{jd}^{k-1}, \forall j \in N^d(i)$ of previous layer from its neighbors $N^d(i)$, followed by nonlinear transformation, which is expressed as,

$$h_{N^d(i)}^k = AGGREGATE_k\left(\left\{h_{jd}^{k-1}, \forall j \in N^d(i)\right\}\right);$$
$$h_{id}^k = \sigma(W^k * CONCAT(h_{id}^{k-1}, h_{N^d(i)}^k))$$

where $W^k$ is the trainable weights and $\sigma$ is the ReLU function. $AGGREGATE_k$ represents $k$th aggregation function, which can be meaning, pooling, or LSTM aggregator. The neighbors $N^d(i)$ of spot $i$ are obtained from $G^d(V^d, E^d)$. We get the final embedding of spot $i$ as,

$$h_{id}^K = W^K CONCAT\left(h_{id}^{K-1}, h_{N^d(i)}^K\right);$$
$$z_{id} = h_{id}^K$$

The loss function of GraphSAGE depicts the ability of learned embeddings to reconstruct original graph, which is expressed as maximizing the similarities of embeddings between nearby nodes and minimizing them between disparate nodes. For embedding $z_{id}$, its loss is described as,

$$l_{gs}(z_{id}) = \sum\nolimits_{j \in RW^d(i)} -\log\left(\rho\left(z_{id}^T z_{jd}\right)\right) - \mathbb{E}_{i_n d \in p_n^d(i)}\log(1 - \rho(z_{id}^T z_{i_n d}))$$

where spot $j$ is a spot that co-occurs with spot $i$ on fixed length random walk, and $RW^d(i)$ represents all possible co-occurrence spots in one minibatch starting at spot $i$ in $G^d(V^d, E^d)$. $\rho$ is the sigmoid function, and $p_n^d(i)$ represents the distribution of negative samples. We treat $(z_{id}, z_{jd})$ as positive pairs and $\left(z_{id}, z_{i_n d}\right)$ as negative pairs. In our implementation, for each minibatch, we sampling almost the same number of positive pairs and negative pairs.

### *Decoder*

The decoder takes the latent embedding as input to reconstruct the processed gene expression. The output of layer $k-1(k = 2, \ldots, K-1)$ is computed by the output of layer $k$ for spot $i$,

$$o_{id}^{k-1} = \sigma(\widehat{W}_k o_{id}^k + \widehat{b}_k)$$

where $\widehat{W}_k$ and $\widehat{b}_k$ are trainable parameters of decoder, and $\sigma$ is the ReLU function. The output of $o_{id}^K$ decoder is as,

$$o_{id}^0 = \rho(\widehat{W}_1 o_{id}^1 + \widehat{b}_1)$$
$$\widehat{x}_{id} = o_{id}^0$$

where $\rho$ is the sigmoid function, and $\widehat{x}_{id}$ is the recovered expression of spot $i$ from sample $d$. The loss function of decoder is formulated as,

$$l_{de}(\widehat{x}_{id}, x_{id}) = \sum_{M} -x_{id}\log(\widehat{x}_{id}) - (1 - x_{id})\log(1 - \widehat{x}_{id})$$

In which the summation operates on $M$ genes.

### Discriminator

The discriminator extracts a smaller dimension of $z_{id}$ as input, labeled as $z_{id}^{bio}$, and prints sample ID as output, labeled as $y_{id}$. The discriminator aims to make batch ID indistinguishable from low-dimensional representation $z_{id}^{bio}$ by teaching generator (GraphSAGE in our model) to fool a well-trained discriminator via adopting gradient reversal layer (GRL) [51]. The output of discriminator can be obtained as,

$$\widetilde{y}_{id}^{bio} = f_{disc}(GRL(z_{id}^{bio}); \widetilde{W}, \widetilde{b})$$
$$GRL(z_{id}^{bio}) = z_{id}^{bio}; \frac{\partial GRL(z_{id}^{bio})}{\partial z_{id}^{bio}} = -1$$

where $f_{disc}$ is a multi-layer nonlinear function, and $\widetilde{W}, \widetilde{b}$ are its trainable parameters. *GRL* functions as gradient flipping with the same input and output in the forward propagation and the opposite input and output in the backward propagation. Note that when the number of batches is two, $y_{id} = 0 \, or \, 1 \, for \, d = 1 \, or \, 2$, and the nonlinear transform of the final layer of $f_{disc}$ is sigmoid function. Otherwise, $y_{id}$ is a zero-vector except the *dth* position is 1 and the final nonlinear function is softmax function. The loss function of the discriminator is,

$$l_{disc}\left(\widetilde{y}_{id}^{bio}, y_{id}\right) = \theta(\widetilde{y}_{id}^{bio}, y_{id})$$

In which $\theta$ is a cross entropy function.

### Classifier

The classifier takes another portion $z_{id}^{noise}$ of $z_{id}$ as input, where $z_{id} = CONCATE(z_{id}^{noise}, z_{id}^{bio})$, and batch ID $y_{id}$ as output. The function of the classifier is to well distinguish batch ID to retain batch information in $z_{id}^{noise}$, whose concatenation with $z_{id}^{bio}$ as $z_{id} = [z_{id}^{noise}, z_{id}^{bio}]$, preserving the original structure, which can reconstruct more accurate original expressions. The design of this classifier is able to obtain gene expressions without batch effects by setting $z_{id}^{noise} = 0, \forall i = 1, \ldots, S, d = 1, \ldots, D$ [14]. The output and loss function of classifier are,

$$\overline{y}_{id}^{noise} = f_{class}(z_{id}^{noise}; \overline{W}, \overline{b})$$
$$l_{class}(\overline{y}_{id}^{noise}, y_{id}) = \theta(\overline{y}_{id}^{noise}, y_{id})$$

where $f_{class}$ is a multi-layer nonlinear function, and $\overline{W}, \overline{b}$ is its trainable parameters. $\overline{y}_{id}^{noise}$, $y_{id}$, and $\theta$ have the same form as the discriminator mentioned before.

### Total loss

The total loss of SPIRAL-integration is weighted summation of the four losses in one minibatch, where $M$ represents the number of genes and

$$l_{total} = \sum_{i,d \in minibatch} l_{gs}\left(z_{id}^{bio}\right) + l_{de}\left(\widehat{x}_{id}, x_{id}\right) * M + l_{disc}\left(\widetilde{y}_{id}^{bio}, y_{id}\right) * \lambda + l_{class}\left(\overline{y}_{id}^{noise}, y_{id}\right) * \gamma$$

### The overall architecture

In all experiments, we set the layer of GraphSAGE as 512–32, the decoder as 32–512, the classifier as 4, and the discriminator as 32–16. The $\lambda$ and $\gamma$ are all set as 1, and the meaning aggregator are used in all experiments. Adam optimizer is used to minimize $l_{total}$, and the initial learning rate and weight decay are set as 0.001 and 0.0005. The batch size is set 16 for simulating data, 256 for sagittal mouse brain, coronal mouse brain and 512 for mouse olfactory bulb and DLPFC, and epoch is set 100 for all experiments. In each minibatch, for each spot, its positive and negative pairs are randomly sampled. We set the number of walks as knn, and the number of steps in one round of walk as 1 for positive pairs. In our experiment, we sampled almost the same number of negative pairs as positive pairs for each spot in minibatch via sampling from spots excluding spots co-occurrence with this spot in random walk.

### Coordinate alignment

Firstly, we selected one of samples as reference, on to which all other samples are aligned. There are no strict conditions for reference selection, which can be the one with complete spatial domains, for example, 10X Visium has larger priority than Slide-seq as it has larger views. Although, do not worry about this problem too much, because we can extend the region of reference coordinate system by aligning new clusters. Next, we registered the shared clusters between the reference sample and the remaining samples using biological embedding. Specifically, for shared cluster $c(c = 1, \ldots, C)$, the spot set of reference sample $R(R \in \{1, \ldots, D\})$ and studied sample $d$ are labeled as $SP_c^R$ and $SP_c^d$, we calculated Gromov-Wasserstein distance like PASTE,

$$F_c\left(\Pi; Z^R, D^R, Z^d, D^d, e, \alpha, c\right) = (1 - \alpha) \sum_{\substack{i \in SP_c^R \\ j \in SP_c^d}} e\left(z_{iR}^{bio}, z_{id}^{bio}\right)\pi_{ij} + \alpha \sum_{\substack{i, k \in SP_c^R \\ j, l \in SP_c^d}} (d_{ik} - d_{jl})^2 \pi_{ij}\pi_{kl}$$

where $e : R^M \times R^M \to R_+$, is a function to calculate the distance of biological embeddings between spots of reference sample and other samples. $\pi_{ij}$ represents the alignment probability of spot $i$ and spot $j$, $d_{ik}$ represents the physical distance of $i$ and spot $k$, the same with $\pi_{kl}$, and $d_{jl}$. $\alpha$ balances the importance between expression profile and spatial structure. Then, we calculated the new coordinates of other samples at reference coordinate system based on $\Pi$. We calculated the new coordinate of spot $j$ of cluster $c$ of sample $d$ as

$$coord_j^R = \frac{1}{\left|\left\{\pi_{ij} > 0, \forall i \in SP_c^R\right\}\right|} \sum_{\pi_{ij} > 0, \forall i \in SP_c^R} coord_i^R$$

where $coord_j^R$ is the new coordinates of spot $j$ in reference coordinate map. Finally, we conducted coordinate transformation, mainly including rotations and translations, to calculate the new coordinates of sample-specific clusters by regarding the spots of shared

clusters as landmark. We used Procrustes function of vegan package of R [52], which rotates a configuration to maximum similarities with another configuration [53, 54].

### Parameter selection

There are two key parameters to be considered, parameter $\lambda$ of SPIRAL-integration, controlling the degree of batch effect removal, and parameter $\alpha$ of SPIRAL-alignment, balancing the importance of gene expressions and spatial correlations. For data from the same tissue sample, $\lambda = 1$ can achieve effective and accurate batch effect removal, while for data having both tissue-shared and significant tissue-specific cell types, such as shared immune cells for many tissues, adjusting $\lambda$ to smaller value would relieve fault mixing of different cell types [14]. For cluster-wise coordination alignment, larger $\alpha$ can compensate for incorrect clustering results by assigning higher weights on gene expressions, and smaller $\alpha$ can preserve spatial correlations more accurately by weighting more on spatial distances. In our studies, $\alpha$ ranges from 0.5 to 0.8 having satisfactory results.

### Clustering methods

For SPIRAL embeddings, we used Louvain algorithm implemented by "scanpy.pp.neighbors" and "scanpy.tl.louvain" of scanpy package on simulated and sagittal mouse brain data, Louvain algorithm implemented by "FindNeighbors" and "FindClusters" of Seurat package on coronal mouse brain and mouse olfactory bulb data, and "mclust_R" of STAGATE package on DLPFC data. For other compared methods, we adopted the provided clustering methods by them.

### Refine clustering

For DLPFC and mouse olfactory bulb data, we refined the cluster label of each spot by its surrounding spots as did in spaGCN [55].

### Metrics description

#### *LISI-CoM*

We evaluated the extent of integrations of same domains and separations of different domains by the local inverse Simpson index (LISI) of batches in each domain (LISI-batch) and the LISI of domains in all data (LISI-domain) [12]. The combination of normalized LISI-batch and LISI-domain is calculated as,

$$\text{LISI\_CoM} = 2 * \frac{(1/LISI\_domain_{norm}) * LISI\_batch_{norm}}{(1/LISI\_domain_{norm}) + LISI\_batch_{norm}}$$

#### *SCS*

We evaluated the spatial coherence of predicted cluster labels on original coordinates and annotated labels on aligned coordinates by z-scaled spatial coherence scores, which was proposed by PASTE [21] by calculating the frequencies of the coexistences of any two labels compared with random assignments of labels,

$$H(G, L) = -\sum_{\{a,b\}:a,b\epsilon K} \mathbb{P}(\{a,b\}|E)\log(\mathbb{P}(\{a,b\}|E))$$
$$\mathbb{P}(\{a,b\}|E) = \frac{n_{\{a,b\}}}{|E|}$$

where $a, b = 1 \ldots K$ represents the labels, and $E$ represents the total number of edges of constructed graphs.

### Moran's I and Geary's C

We evaluated the spatial autocorrelations by Moran's *I* and Geary's *C*, which are used to measure spatial autocorrelations of genes in SRT data analysis [37, 38]. We calculated these two values by two R functions, "Moran.I" of package "ape" and "geary.test" of package "spdep." The values of Geary's *C* are converted by subtracting by 1 to make sure lager value means lager spatial autocorrelations.

### Pseudo time

The pseudo-time is calculated by sc.tl.dpt of "scanpy" package, which is used to calculate spearman correlations between laminar labels from 1 to 7, representing from L1 to WM.

### Data description

### Simulating data

We adopted Splatter to simulate gene expressions. We simulated four two-batch datasets and one three-batch dataset, all of which contain 500 genes. In simulate1-simulate3, we simulated the same cell type compositions and two similar cell types: Group1 and Group2; in simulate4 and simulate 5, we generated different cell type compositions. For spatial positions, we randomly assigned cells of each cell type to a restricted regular region. In simulate1, the spatial distributions of cell types are the same; in simulate2, the spatial ordering of cell types among two batches are reversed; and in simulate3, the coordinates of one batch are rotated on the basis of simulate2. In simulate4 and simulate5, we assumed the same spatial arrangements of shared cell types.

### DLPFC data

The DLPFC data consists of 12 sections for three donors, each donor has two pairs of $10\mu$m replicates, and the two pairs of replicates are $30\mu$m apart. The number of spots ranges from 3498 to 4789 for each sample, and the layers were manually labeled, which are ground truth for comparison.

### Sagittal mouse brain data

The sagittal mouse brain is divided along the anterior-posterior axis, which have one pair of replicates respectively. The number of spots ranges from 2696 to 3353. We annotated domains by domain-marker genes and H&E staining images. The links of these data are as follows:

https://support.10xgenomics.com/spatial-gene-expression/datasets/1.0.0/V1_Mouse_Brain_Sagittal_Anterior

https://support.10xgenomics.com/spatial-gene-expression/datasets/1.0.0/V1_Mouse_Brain_Sagittal_Anterior_Section_2

https://support.10xgenomics.com/spatial-gene-expression/datasets/1.0.0/V1_Mouse_Brain_Sagittal_Posterior

https://support.10xgenomics.com/spatial-gene-expression/datasets/1.0.0/V1_Mouse_Brain_Sagittal_Posterior_Section_2

### Coronal mouse brain data

There three coronal mouse brain sections, which are formalin-fixed paraffin-embedded (FFPE) tissue with H&E staining, fresh frozen tissue with H&E staining, and fresh frozen tissue with immunofluorescence (IF) staining. The number of spots is 2264, 2702, 2903. We labeled these tissues by marker genes and Allen brain atlas.

The links of these data are as follows:

https://www.10xgenomics.com/resources/datasets/mouse-brain-section-coronal-1-standard-1-1-0

https://www.10xgenomics.com/resources/datasets/adult-mouse-brain-section-1-coronal-stains-dapi-anti-neu-n-1-standard-1-1-0

https://www.10xgenomics.com/resources/datasets/adult-mouse-brain-ffpe-1-standard-1-3-0

### Mouse olfactory bulb data

We used mouse olfactory bulb data of 10X Visium, Stereo-seq and Slide-seq V2, whose resolutions are $50\mu$m, $\sim 35\mu$m, and $10\mu$m, and the number of spots is 1185, 8827, 18,537. We used Seurat to preprocess the Slide V2 data, which includes calculating and plotting the quality control features and filtering the low-quality spots, that with less than 3000 UMIs and 15 genes. For Stereo-seq data, we used the in-house built R program to merge transcripts to bins, the detail process includes specify a bin_size, adjust coordinates, calculate bin IDs, aggregate UMI counts, and get the new coordinates of each bin. In this paper, we used bin_size as 50 to obtain approximately 35μm resolution. The 10X Visium data can be accessed via https://www.10xgenomics.com/resources/datasets/adult-mouse-olfactory-bulb-1-standard.

## Supplementary Information

**Additional file 1: Figure S1.** Simulating. A. Spatial arrangements of cell types on original and aligned coordinates by SPIRAL and PASTE for simulate 1-3 datasets. Colors represent cell types and shapes represent batches. B-C UMAP visualizations of simulate2-3 (B) and simulate4-5 (C) datasets, each dataset occupies two rows, in which colors represent cell types (upper) and batches (bottom). **Figure S2.** DLPFC. A. UMAP visualizations of four-section integrations for each donor (donor1 in upper, donor2 in middle, donor3 in bottom). B. Spatial arrangements of annotated layers on aligned coordinates by SPIRAL and PASTE (left, middle) and spatial distributions of clusters (mclust; right) on SPIRAL-aligned coordinates. C. Spatial distributions of raw, Seurat and SPIRAL-integrated domain-marker gene expressions, RORB, KRT17 and PCP4 in 151507, 151669, 151673 and SPIRAL-aligned coordinates. **Figure S3.** Sagittal mouse brain. A. H&E staining images of both anterior and posterior mouse brains. B. The anatomy diagram of sagittal mouse brain, where black line divides mouse brain into anterior and posterior parts. C. Spatial visualizations of annotated domain labels for four samples. D. UMAP visualizations of raw and integrated embeddings by five methods in

integrating four samples. Each method occupies two columns, where domains are colored in left and batches are colored in right. E. Spatial arrangements of annotated domains on samples with one of replicates were rotated 60 degree. F. Spatial arrangements of annotated domains on aligned coordinates of anterior1&2 (left) and posterior1&2 (right) by SPIRAL (upper) and PASTE (bottom). G. Spatial disposition of mclust-derived clusters by SPIRAL on four samples. H. Comparison of the expressions of DEGs from cluster10 and DEGs from cluster 2,3,15,21 in mouse olfactory bulb domains. **Figure S4.** Coronal mouse brain. A. Annotations of domains generated by marker genes and Allen mouse brain atlas in three coordinates. B. UMAP visualizations of integrations of embeddings from origin and five methods. C. The anatomy diagram of coronal mouse brain for hippocampus structures (left) and the distributions of the corresponding gene expressions (right). D. The arrangements of derived clusters by for spatial based methods on the region of hippocampus. E. Statistics of sequence depth of three protocols. F. Spatial arrangements of annotated domains on aligned coordinates by SPIRAL (left) and PASTE (right). G. Spatial patterns of domain-marker gene expressions from raw data and SPIRAL integrated data in the cortex region of 10X Normal data. **Figure S5.** Mouse olfactory bulb. A. Spatial distributions of clusters derived by SPIRAL in aligned coordinates by SPIRAL (left) and PASTE right). B&C. Spatial distributions of Louvain-derived clusters on the embeddings of harmony_STAGATE (B) and STAligner (C). D. Spatial distributions of predicted clusters of 10X Visium data using model trained by Stereo seq data and Slide V2 data. E. Spatial distributions of of predicted clusters on aligned coordinates by SPIRAL. F. The percentages of cell types in each cluster. G. Spatial autocorrelations of raw and integrated gene expressions in original and aligned coordinates. H. Spatial distributions of raw, SPIRAL-integrated and Seurat-integrated layer-marker expressions, Nrgn, Gabra1, Slc6a11 and Kctd12 on three samples. Each method occupies one row.

**Additional file 2.** Supplementary Table.

**Additional file 3.** Review History.

## Acknowledgements
The authors want to thank Minping Qian, Juntao Gao, and Minglei Shi for their comments and suggestions; Honjun Li for his technical support; and Zhenyi Wang for the discussions.

## Peer review information

## Review history
The review history is available as Additional file 3.

## Authors' contributions
T.T.G, Z.Y.Y, M.Q.Z, and X.Y.L initiated the project. T.T.G, Z.Y.Y, and X.Y.L developed the method. T.T.G, Z.Y.Y, J.K.W, and F.L.C performed the data analysis. Y. P collected and annotated the original data. T.T.G, Z.Y.Y, M.Q.Z, and X.Y.L wrote the manuscripts. All authors read and approved the final manuscript.

## Funding
This study was supported by National Natural Science Foundation of China (62003028). X.L. was supported by a Scholarship from the China Scholarship Council. This work was also supported by Shanghai Science and Technology Development Funds (23YF1403000), Chenguang Program of Shanghai Education Development Foundation and Shanghai Municipal Education Commission (22CGA02), Tencent AI Lab Rhino-Bird Focused Research Program (RBFR2023008), Shanghai Center for Brain Science and Brain-Inspired Technology, and 111 Project (No.B18015).

## Availability of data and materials
SPIRAL is implemented as a python tool under the GNU General Public License v3.0. The reproducible codes and all data used in this paper are available at GitHub [56] and Zenodo [57].

# Declarations

## Ethics approval and consent to participate
No ethical approval was required for this study. All utilized public datasets were generated by other organizations that obtained ethical approval.

## Competing interests
The authors declare no competing interests.

## Author details
[1]School of Software Engineering, Beijing Jiaotong University, Beijing 100044, China. [2]MOE Key Laboratory of Bioinformatics, Bioinformatics Division and Center for Synthetic & Systems Biology, BNRist, Department of Automation, Tsinghua University, Beijing 100084, China. [3]Institute of Science and Technology for Brain-Inspired Intelligence, Center for Medical Research and Innovation, Shanghai Pudong Hospital, Fudan University Pudong Medical Center, MOE Key Laboratory of Computational Neuroscience and Brain-Inspired Intelligence, Fudan University, Shanghai 200433, China. [4]School of Biomedical Sciences, LKS Faculty of Medicine, The University of Hong Kong, Hong Kong SAR, China. [5]Center for Stem Cell Biology and Regenerative Medicine, MOE Key Laboratory of Bioinformatics, School of Life Sciences, Tsinghua-Peking Center for Life Sciences, Tsinghua University, Beijing 100084, China. [6]Department of Biological Sciences, Center for Systems Biology, The University of Texas, Richardson, TX 75080-3021, USA.

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
