## [**Additional file 3.** Review History. · Genome Biology]

Review History

First round of review

Reviewer 1

Were you able to assess all statistics in the manuscript, including the appropriateness of statistical tests used? Yes.

Were you able to directly test the methods? No.

Comments to author:

Guo et al. report a new computational framework, SPIRAL, that can be employed for integration and alignment of spatial transcriptomics (ST) datasets. As ST technologies have starkly increase in popularity in the last years, it is essential to have computational methods that can efficiently integrate datasets stemming from different technologies, as well as align multiple tissue sections.

In this regard, SPIRAL holds the potential to be a vital tool within the ST field. The work methodology is clearly presented and the manuscript is overall well-written, with the exception of a few typos that can be caught and corrected during proofreading.

SPIRAL consists of two modules that offer integration and alignment of ST data. The authors showcase SPIRAL's capabilities in synthetic and published datasets spanning different technologies (10X Visium, Slide-seq, Stereo-seq). SPIRAL seems to outperform existing methods that were originally developed for scRNA-seq (Seurat & Harmony), as well as PASTE, a method developed for the analysis of ST data.

Overall, the work is of high quality and I would strongly recommend it for publication, provided that the authors address the following major point. Several computational methods have been developed for analysis of ST data, including alignment of different sections. The authors should discuss these methods and benchmark SPIRAL against existing ones wherever appropriate (e.g. ST Viewer, STIM & STUtility etc. for alignment; squidpy, Giotto stLearn & DeepST for analysis & integration etc.)

Along these lines, the manuscript would greatly benefit by a more extensive discussion of SPIRAL's limitations. Expanding on the last paragraph of discussion: what are SPIRAL's boundaries and in which cases is SPIRAL expected to underperform?

Reviewer 2

Were you able to assess all statistics in the manuscript, including the appropriateness of statistical tests used? Yes.

Were you able to directly test the methods? No.

Comments to author:

In this manuscript, the authors proposed a spatial integration and alignment method for spatially resolved transcriptomics data. SPIRAL uses graph neural network and domain adaptation to learn the batch-corrected embeddings and expressions, and optimal transportation to achieve slice coordination alignment. Although SPIRAL may be a useful tool, its reproducibility is still insufficient and benchmarking is not rigorous. My main concerns and comments that should be addressed are as follows.

Major Comments:

- The novelty of the algorithm is limited. The combination of graph neural network and domain adaptation to achieve spatially aware integration of SRT data is not a new idea, which has been used in previous work DeepST for integrating multiple SRT slices (<https://doi.org/10.1093/nar/gkac901>). Also, several methods have been developed to address very related issues (e.g., <https://doi.org/10.1186/s13059-022-02734-7>, <https://doi.org/10.1038/s41467-023-36796-3>, <https://doi.org/10.1101/2022.12.26.521888>). The authors are supposed to clarify the originality of this work and compared it with these highly related studies.

- In the model, how to balance the importance of the architecture Discriminator, Classifier, and GraphSAGE? Additional ablation studies are supposed to demonstrate the necessity of the three architectures.

- The tutorials provided in GitHub are unclear and not easy to reproduce the main results. For example, at the end of https://github.com/guott15/SPIRAL/blob/main/Demo/run_spiral_saggital.ipynb, the sagittal brain slices are not aligned by SPIRAL, which is inconsistent with the results in Figure 4. The `knn_classify.ipynb` demonstrates an error message and no results are reported. These problems may make the users confused about the reliability of SPIRAL. The authors are supposed to fix these bugs and provide step-to-step tutorials on how to use SPIRAL.

- It seems that the original coordinates are well aligned in Figure 4. Is it necessary to align the coordinates for anterior and posterior replicates? At least, the authors may need to initialize the orientation randomly.

- It seems that the authors mix the concept of spatial domain and cell type. Actually, for sagittal mouse brain and coronal mouse brain data (Fig.4-5, Table S1), the "domains" annotated by the authors are cell types and not true spatial tissue structures. Additionally, the annotation provided in Fig. S3C and Fig. S5A by the authors are not accurate and the finer structures such as rostral migratory stream (RMS) are missed. Inaccurate annotations are not suitable to be used in

benchmarking. Besides the domain markers in Table S1, visualization of their spatial expression is also supposed to provide in supporting the domain annotation.

- The statement "It is striking that the finer structure of RMS, which would not have been found by sole analysis of the low-resolution data (Figure S5A), was revealed by joint clustering (cluster 7 of Figure 6D)." seems not appropriate. The finer structure of RMS has been found by sole analysis with many published spatial domain identification tools, such as DeepST, STAGATE, SpaceFlow.

- I think only comparison to STAGATE but no other spatial clustering and embedding methods are not valid and sufficient. Even so, I doubt whether the authors well re-implemented it. The performance like Figures 3, 4, 5, and Figure S5C are invalid, and some are not consistent with the original one in STAGATE (<https://stagate.readthedocs.io/en/latest/AT2.html>). The authors may need to double-check if they correctly re-used this method. To make the comparison fair, the authors are supposed to re-run the analysis of harmony_STAGATE and provide the details of running competing methods. Anyway, the benchmarking is still in its stage.

- Some analysis and demonstration can be improved. For example, Figure 6A is a bit redundant with two replicated subplots, and too much blank in the third. The plot of Slide-seq V2 in Figure 6D is problematic, maybe there are some noisy parts (not real tissue) therein that should be filtered.

Minor Comments:

- It seems to me all the supplementary figures do not have captions.

Dear Editor,

Thanks for your and the reviewers' time and efforts in reviewing our manuscript. The comments and suggestions have greatly helped us to improve the manuscript. We have carefully revised the manuscript, and the major four points are summarized below:

1. We added the comparison of SPIRAL-integration with eight state-of-the-art methods and SPIRAL-alignment with two methods, using both simulated and published spatially resolved transcriptome datasets. Overall, SPIRAL demonstrates excellent performance across various classes of batch effects, including diverse slices, experimental protocols and technologies.
2. We did ablation studies to measure the critical role of each SPIRAL component in contributing to its superior performance.
3. To ensure reproducibility, we have shared a detailed tutorial and all analysis code used in this study on Github: <https://github.com/guott15/SPIRAL>. This includes detailed explanations of our annotations for data without a ground truth and execution codes for STAGATE and other methods.
4. We carefully checked and revised all the proposed language problems to improve its clarity.

Please see the following point-by-point reply to the reviewers.

Yours sincerely,
Xiangyu Li

Reply to Referee: 1

Comments for the Author

Guo et al. report a new computational framework, SPIRAL, that can be employed for integration and alignment of spatial transcriptomics (ST) datasets. As ST technologies have starkly increase in popularity in the last years, it is essential to have computational methods that can efficiently integrate datasets stemming from different technologies, as well as align multiple tissue sections. In this regard, SPIRAL holds the potential to be a vital tool within the ST field. The work methodology is clearly presented and the manuscript is overall well-written, with the exception of a few typos that can be caught and corrected during proofreading.

SPIRAL consists of two modules that offer integration and alignment of ST data. The authors showcase SPIRAL's capabilities in synthetic and published datasets spanning different technologies (10X Visium, Slide-seq, Stereo-seq). SPIRAL seems to outperform existing methods that were originally developed for scRNA-seq (Seurat & Harmony), as well as PASTE, a method developed for the analysis of ST data.

Reply: We thank the reviewer for the positive evaluation and constructive suggestions, which have greatly improved the quality of the manuscript. Our detailed responses to your specific comments are listed below.

1. Overall, the work is of high quality and I would strongly recommend it for publication, provided that the authors address the following major point. Several computational methods have been developed for analysis of ST data, including alignment of different sections. The authors should discuss these methods and benchmark SPIRAL against existing ones wherever appropriate (e.g. ST Viewer, STIM & STUtility etc. for alignment; squidpy, Giotto stLearn & DeepST for analysis & integration etc.)

Reply: We thank the reviewer for this comment. Following your suggestions, we have added comprehensive comparisons between SPIRAL and eight methods (Seurat, harmony, harmony_STAGATE, harmony_SEDR, GraphST, DeepST, BASS, STAligner) into our revised manuscript. These comparisons are presented in Figure 2, 3, 4, 5 and 6 for reference. Furthermore, we have enhanced the introduction section with additional discussion.

For alignments methods, we compared SPIRAL-alignment, PASTE and STIM using spatial coherence score [1] on DLPFC data to align four slices of each donor (refer to Figure A below). STUtility and ST Viewer were not included in the comparative analysis with SPIRAL-alignment for the following reasons: 1) STUtility aligns images without considering gene expressions or spot locations, leading to the lack of the mapping information of the aligned spots, which further limits its ability to perform SCS calculations and other downstream tasks [1-2]. Additionally, STUtility and STIM can integrate adjacent slices of one tissue, but are ineffective for slices with considerable differences in spatial structure and cell composition [2-3]. 2) ST Viewer provides real-time interaction, analysis, visualization of Spatially Transcriptomics (ST) datasets, using 2D or aligned 3D slices as inputs. However, it lacks the functionality to align distinct slices [4].

The results in Figure A demonstrate the superiority of SPIRAL in aligning the more diverse slices of 'Donor1' and 'Donor3', compared to 'Donor 2'. In experiments aimed at aligning 12 slices across multiple donors and three coronal mouse brain samples with different protocols, STIM fails to produce any results.

Performance comparisons of three alignment methods by the metric of SCS on DLPFC data

In our comparison of integration methods, we included other spatial-based methods with multi-sample analysis function, namely DeepST, STAligner [5], GraphST [6] and BASS [7]. Squidpy, Giotto and stLearn were excluded from the comparison due to their specific functionalities:

- 1) Squidpy focuses on integrating transcriptome and morphology for analyzing spatial transcriptome or protein data, enabling tasks such as nuclei segmentation, neighborhood analysis and ligand-receptor interaction analysis, but only for a single sample [8];
- 2) Giotto can analyze and visualize single-cell spatial expression data with diverse resolutions, but again, it is designed for analysis within a single sample [9];
- 3) stLearn is designed to comprehensively analyze ST data by combining spatial distance, tissue morphology and gene expression, but it also operates on a single sample [10].

To assess batch effect correction and spatial coherence preservation, we utilized four metrics: LISI-CoM, iLISI, ARI and SCS. It is noted that, for BASS, the lack of integrated embedding and gene expression prevented the calculation of LISI-CoM and iLISI. Our results demonstrated that SPIRAL-integration consistently exhibited superior performance.

Performance comparisons of nine methods for four metrics on four datasets. (A) displays LISI-CoM for 13 integration experiments on DLPFC dataset, (B) displays iLISI on sagittal, coronal and olfactory bulb datasets, (3B) displays ARI by mclust method on 13 integration experiments on DLPFC dataset (C) displays SCS on the DLPFC, sagittal, coronal and olfactory bulb datasets.

In summary, our results reveal the superior performance of SPIRAL over other methods, and we hope these results can clarify your concerns.

2. Along these lines, the manuscript would greatly benefit by a more extensive discussion of SPIRAL's limitations. Expanding on the last paragraph of discussion: what are SPIRAL's boundaries and in which cases is SPIRAL expected to underperform?

Reply: We thank the reviewer for the valuable comment. Following your suggestion, we now have included a more comprehensive discussion in the revised manuscript (Page 12), which reads as follows:

While SPIRAL has demonstrated effectiveness, there is still room for improvement. Firstly, SPIRAL could further combine histological image, spatial locations and gene expressions to remove batch effects both in gene expressions and in images, aligning both spots and pixels. This would enhance the delineation of morphological structures across multiple samples. Secondly, SPIRAL-alignment could be improved to ensure a more evenly distributed alignment within each cluster. Thirdly, while SPIRAL performs well with current datasets, its utility could be extended to higher resolution data, such as MERFISH [11] data, to achieve more precise delineation of structures. Additionally, parallel computation and distributed learning could be utilized to accelerate training procedures [12-13]. Moreover, as spatial multi-omics data become increasingly available, integration strategies across various spatial omics would be desperately needed. SPIRAL could expand its capabilities to integrate a variety of spatial multimodal data, such as FISH-based SRT data and spatial proteomics data [14-15]. This expansion holds great promise for gaining a deeper understanding of cell heterogeneity and complex tissue organization.

Reference

- [1] Zeira R, Land M, Strzalkowski A, et al. Alignment and integration of spatial transcriptomics data[J]. *Nature Methods*, 2022, 19(5): 567-575.
- [2] Bergenstråhle J, Larsson L, Lundeberg J. Seamless integration of image and molecular analysis for spatial transcriptomics workflows[J]. *BMC genomics*, 2020, 21(1): 1-7.
- [3] Preibisch S, Karaiskos N, Rajewsky N. Image-based representation of massive spatial transcriptomics datasets[J]. *bioRxiv*, 2021: 2021.12.07.471629.
- [4] Fernández Navarro J, Lundeberg J, Ståhl P L. ST viewer: a tool for analysis and visualization of spatial transcriptomics datasets[J]. *Bioinformatics*, 2019.
- [5] Zhou X, Dong K, Zhang S. Integrating spatial transcriptomics data across different conditions, technologies, and developmental stages[J]. *bioRxiv*, 2022: 2022.12.26.521888.
- [6] Long Y, Ang K S, Li M, et al. Spatially informed clustering, integration, and deconvolution of spatial transcriptomics with GraphST[J]. *Nature Communications*, 2023, 14(1): 1155.
- [7] Li Z, Zhou X. BASS: multi-scale and multi-sample analysis enables accurate cell type clustering and spatial domain detection in spatial transcriptomic studies[J]. *Genome biology*, 2022, 23(1): 168.
- [8] Palla G, Spitzer H, Klein M, et al. Squidpy: a scalable framework for spatial omics analysis[J]. *Nature methods*, 2022, 19(2): 171-178.
- [9] Dries R, Zhu Q, Dong R, et al. Giotto: a toolbox for integrative analysis and visualization of spatial expression data[J]. *Genome biology*, 2021, 22: 1-31.
- [10] Pham D, Tan X, Xu J, et al. stLearn: integrating spatial location, tissue morphology and gene expression to find cell types, cell-cell interactions and spatial trajectories within undissociated tissues[J]. *BioRxiv*, 2020: 2020.05.31.125658.
- [11] Zhang, M., Eichhorn, S. W., Zingg, B., Yao, Z., Cotter, K., Zeng, H., ... & Zhuang, X. (2021). Spatially resolved cell atlas of the mouse primary motor cortex by MERFISH. *Nature*, 598(7879), 137-143
- [12] Li, S., Zhao, Y., Varma, R., Salpekar, O., Noordhuis, P., Li, T., ... & Chintala, S. (2020). Pytorch distributed: Experiences on accelerating data parallel training. *arXiv preprint arXiv:2006.15704*.
- [13] Fey, M., & Lenssen, J. E. (2019). Fast graph representation learning with PyTorch Geometric. *arXiv preprint arXiv:1903.02428*.
- [14] Gut, G., Herrmann, M. D., & Pelkmans, L. (2018). Multiplexed protein maps link subcellular organization to cellular states. *Science*, 361(6401), eaar7042.
- [15] Goltsev, Y., Samusik, N., Kennedy-Darling, J., Bhate, S., Hale, M., Vazquez, G., ... & Nolan, G. P. (2018). Deep profiling of mouse splenic architecture with CODEX multiplexed imaging. *Cell*, 174(4), 968-981.

Reply to Referee: 2

Comments for the Author

In this manuscript, the authors proposed a spatial integration and alignment method for spatially resolved transcriptomics data. SPIRAL uses graph neural network and domain adaptation to learn the batch-corrected embeddings and expressions, and optimal transportation to achieve slice coordination alignment. Although SPIRAL may be a useful tool, its reproducibility is still insufficient and benchmarking is not rigorous. My main concerns and comments that should be addressed are as follows.

Reply: We thank the reviewer for the valuable time and efforts in reviewing our manuscript, and for the detailed suggestions provided. We now have incorporated additional analyses and discussions to address all concerns. Please find our point-by-point responses below.

Major Comments for the Author

1. The novelty of the algorithm is limited. The combination of graph neural network and domain adaptation to achieve spatially aware integration of SRT data is not a new idea, which has been used in previous work DeepST for integrating multiple SRT slices (<https://doi.org/10.1093/nar/gkac901>). Also, several methods have been developed to address very related issues (e.g., <https://doi.org/10.1186/s13059-022-02734-7>, <https://doi.org/10.1038/s41467-023-36796-3>, <https://doi.org/10.1101/2022.12.26.521888>). The authors are supposed to clarify the originality of this work and compared it with these highly related studies.

Reply: We thank the reviewer for raising this question. We agree with the reviewer that the domain adaptation strategy has been employed for integrating scRNA-seq or SRT data before, such as our previous model SCIDRL [1] (published in 2022.01) and DeepST (published 2 months after the submission of our manuscript). However, our method introduces several key advancements and novelties compared to the existing methods.

- 1) SPIRAL accomplishes simultaneous spatial coherence preservation and batch effect correction for spatial transcriptomics data, while our previous model SCIDRL targeted scRNA-seq data.
- 2) SPIRAL-integration generates both batch-corrected embeddings and gene expressions using a noise classifier, enabling a broad range of analyses at both the cell-level and gene-level for biological discovery. In contrast, DeepST, GraphST and STAligner corrects the batch effects in the embedding space, suitable for cell-level analysis like domain detection, but not applicable for gene-level analysis.
- 3) SPIRAL-integration employs GraphSAGE, an inductive method, instead of the transductive graph convolutional network used in DeepST, enabling more flexible implementation of new spots or new slices.
- 4) SPIRAL-integration adopts a minibatch training mode. This approach alleviates memory constraints in large-scale networks and facilitates the integration of diverse samples, such as integrations of coronal mouse brain by three protocols (Figure S4A in revised manuscript) and mouse olfactory bulb samples by two technologies (Figure A in below). In contrast to DeepST, where all spots were trained at one time, our method allows for more effective training.
- 5) For the remaining methods, some limitations existed. For example, GraphST heavily relies on the PASTE method, and BASS lacks integrated embedding and gene expression. A

comprehensive analysis of these methods' limitation is detailed in the 'Introduction' of our revised manuscript (Page 3 of revised manuscript).

Performance comparisons of SPIRAL and DeepST on two datasets. **(S4B)** UAMP visualizations for coronal mouse brain SRT data, different colors represent different protocols, **(A)** UAMP visualizations for mouse olfactory bulb SRT data, different colors represent different technologies.

Following your suggestions, we have added DeepST, GraphST, BASS and STAligner into our benchmark methods across all experiments. To evaluate the performance of SPIRAL, we used visualizations of the UMAP representation and the spatial arrangements of derived clusters, along with quantitative metrics including LISI-CoM or iLISI, ARI and SCS. These metrics assess batch mixtures, joint clustering accuracy, and spatial coherence preservation. To provide a convenient and intuitive summary of our results, we have summarized the evaluation performance across datasets in Figure B, C&D below, and Figure 3B of the revised manuscript. SPIRAL-integration achieved the highest LISI and ARI among all datasets and relatively high SCS. In summary, SPIRAL-integration demonstrated better spatial dependency retaining compared to single-cell based methods, and a higher degree of batch effect removal compared to spatial based methods.

Performance comparisons of nine methods for four metrics on four datasets. **(B)** displays LISI-CoM for 13 integration experiments on the DLPFC dataset, **(C)** displays iLISI on the sagittal, coronal and olfactory bulb datasets, **(3B)** displays ARI by the mclust method on 13 integration experiments on the DLPFC dataset **(D)** displays SCS on the DLPFC, sagittal, coronal and olfactory bulb datasets.

2. In the model, how to balance the importance of the architecture Discriminator, Classifier, and GraphSAGE? Additional ablation studies are supposed to demonstrate the necessity of the three architectures.

Reply: We thank the reviewer for this question and suggestion. In SPIRAL-integration,

GraphSAGE functions as an encoder, learning the embeddings of each spot and aggregating the information among neighbors. The biology discriminator and the noise classifier are key structures to achieve disentanglement. Through the disentanglement process of SPIRAL, the original embeddings obtained by GraphSAGE can be transformed to two parts: one representing a batch-effect-invariant biological embedding and the other reflecting batch discrepancy. To access the impact of each structure, we did ablation studies on the 151507 and 151673 slices of the DLPFC SRT data by removing each component from SPIRAL. SPIRAL with GraphSAGE achieves significantly higher SCS (Figure A below) and ARI (Figure B below) scores than SPIRAL without GraphSAGE, indicating the importance of GraphSAGE in preserving spatial coherence. SPIRAL with a discriminator results in significantly higher LISI-CoM scores compared to SPIRAL without a discriminator (Figure C below), highlighting the essential function of biology discriminator in effectively removing batch effect. Furthermore, to intuitively evaluate the effectiveness of the noise classifier, we visualized the recovered gene expressions that were learnt with and without the noise classifier (Figure D&E below). The gene expression space without the noise classifier exhibited more explicit batch effects, providing evidence for the effectiveness of the noise classifier. In summary, each component of SPIRAL contributes to the successful removal of batch effects and the retention of spatial coherence.

Performance comparisons of SPIRAL-integration model and models without specific structures on DLPFC data. (A&B) display the ARI and SCS of SPIRAL and SPIRAL without GraphSAGE, respectively (C) displays the LISI-CoM of SPIRAL and SPIRAL without a discriminator, (D&E) UMAP visualizations of the recovered gene expressions by SPIRAL with (D) and without (E) the noise classifier.

3. The tutorials provided in GitHub are unclear and not easy to reproduce the main results. For example, at the end of https://github.com/guott15/SPIRAL/blob/main/Demo/run_spiral_saggital.ipynb, the sagittal brain slices are not aligned by SPIRAL, which is inconsistent with the results in Figure 4. The `knn_classify.ipynb` demonstrates an error message and no results are reported. These problems may make the users confused about the reliability of SPIRAL. The authors are supposed to fix these bugs and provide step-to-step tutorials on how to use SPIRAL.

Reply: Thank you very much for raising this question. We sincerely apologize for any confusion caused by the unclear description. For clarification, we have compiled a comprehensive tutorial which includes relevant examples. The SPIRAL code, accessible via Github (<https://github.com/guott15/SPIRAL/>), provides several 'ipynb' files in the 'Demo' folder. Each file is tailored for a specific dataset and guides the execution of sequential functions such as spiral integration, clustering, smooth clustering, spiral alignment, spiral prediction, and knn prediction. We have run the 'run_spiral_saggital.ipynb' notebook anew, resulting in accessible spatial visualizations of aligned coordinates (as seen in the Figure below and on Github). We've also included the 'knn prediction' section in the 'run_spiral_mouse_OB.ipynb' notebook and have

displayed the results therein.

```

Step1:SPIRAL integration

feat:ncell * ngene matrix

edge:edge * 2: obtained from GenerateEdges.ipynb

meta:ncell * 1或ncell * 2,containing batch information, columns is 'batch'

step2: clustering

import anndata
import scanpy as sc
ann=anndata.AnnData(SPIII.feat)
ann.obs['spiral']=embed1.loc[:,SPIII.params.znoise_dim:]
sc.pp.neighbors(ann,use_rep='spiral')
sc.tl.leiden(ann,resolution=1.2)
sc.tl.louvain(ann,resolution=1.2)
ann.obs['batch']=SPIII.meta.loc[:, 'batch']
ub=np.unique(ann.obs['batch'])
sc.tl.umap(ann)
coord=pd.read_csv(coord_file[0],header=0,index_col=0)
for i in np.arange(1,len(samples)):
    coord=pd.concat((coord,pd.read_csv(coord_file[i],header=0,index_col=0))

coord.columns=['x','y']
ann.obs['spatial']=coord.loc[ann.obs_names,:]
cluster_file=dirs+'stt_output/SPIRAL'+flags+'_louvain.csv'
pd.DataFrame(ann.obs['louvain']).to_csv(cluster_file)

Step3:SPIRAL alignment

9): cluster_name='seuratmethod'
input_file=[meta_file,coord_file,embed_file,cluster_file]
output_dirs=dirs+'stt_output/SPIRAL_alignment/'
if not os.path.exists(output_dirs):
    os.makedirs(output_dirs)
ub=['10X_Normal','10X_BMT','10X_PFFP']

0): alpha=0.5
type='weighted_mean'
cluster_name='seurat'
R_dirs='home/guo/tpu2/miniconda3/envs/stnet/lib/R'
CA=CoordAlignment(input_file=input_file,output_dirs=output_dirs,ub=ub,flags=flags,cluster_name=cluster_name,R_dirs=R_dirs,alpha=alpha,type=type)
New_Coord=CA.New_Coord
New_Coord.to_csv(output_dirs+'new_coord'+flags+'_modfy.csv')
adata.obs['aligned_spatial']=New_Coord.loc[adata.obs_names,:].values
adata.obs['celltype']=SPIII.meta.loc[:, 'celltype']

```

Procedures in sagittal mouse brain SRT data

4. It seems that the original coordinates are well aligned in Figure 4. Is it necessary to align the coordinates for anterior and posterior replicates? At least, the authors may need to initialize the orientation randomly.

Reply: Thank you for raising this question. To further show the alignment performance of SPIRAL, we applied 30, 60, 90, 120, 150 and 180-degree rotations to one of the replicates of both posterior and anterior samples (the 60-degree rotation was shown in Figure 3SE in the revised manuscript). Both the boxplots of the SCS for the six categories of rotations and the spatial arrangements of annotated domains of the 60-degree rotation indicate that SPIRAL-alignment and PASTE exhibit comparably satisfactory results.

Performance comparisons of SPIRAL-alignment and PASTE on sagittal mouse brain samples with one replicate rotated certain degrees. (S3E) displays the rotated coordinates, with different colors represent different domains.

(4C) displays the SCS metric for both methods used in aligning either anterior or posterior samples, (S3F) displays spatial arrangements of domains in the aligned coordinates produced by two methods.

5. It seems that the authors mix the concept of spatial domain and cell type. Actually, for sagittal mouse brain and coronal mouse brain data (Fig.4-5, Table S1), the "domains" annotated by the authors are cell types and not true spatial tissue structures. Additionally, the annotation provided in Fig. S3C and Fig. S5A by the authors are not accurate and the finer structures such as rostral migratory stream (RMS) are missed. Inaccurate annotations are not suitable to be used in benchmarking. Besides the domain markers in Table S1, visualization of their spatial expression is also supposed to provide in supporting the domain annotation.

Reply: We thank the reviewer for this question and suggestion. Regarding unannotated SRT data, such as sagittal and coronal mouse brain samples, we initially applied Seurat pipeline [2] for clustering, followed by cluster annotation guided by anatomical structures and domain/layer marker genes (refer to Table S1 of our revised manuscript). These marker genes were carefully selected from reputable literature or databases to ensure their credibility.

When annotating sagittal and coronal mouse brain SRT data (Figure 4,5&S3C in our revised manuscript), both domain and cell type annotations were employed simultaneously due to the constraints in marker information for certain domains. For instance, instead of using the 'L1' of cortex, we annotated the outermost layer utilizing the 'Astrocyte' cell type owing to the scarcity of L1 Layer's marker genes [3]. Given that L1 is predominantly composed of astrocytes in terms of cell bodies [4], we believe it is more rigorous to annotate this outermost cluster as 'Astrocyte' cell type rather than as Layer 1. This annotation process involves a carefully manually examination, performed cluster by cluster. Actually, the evaluation of SPIRAL's performance remains unaffected by whether certain clusters are annotated based on cell type or spatial domain.

In the higher-resolution mouse olfactory bulb data, we initially applied coarser annotations to avoid irregularities of the annotated domains, which inadvertently led to the omission of the RMS structure (refer to Figure S5A in our original manuscript). Upon realizing that the annotation approach mentioned above, while effective for sagittal and coronal mouse brain SRT data, did not accurately capture the finer structures in the mouse olfactory bulb datasets. Therefore, we have deprecated the initial annotations in Figure S5A in our revised manuscript. Now, we have replaced LISI-CoM with iLISI for better clarity (Figure 6B in our revised manuscript), as iLISI is independent of domain annotation. We have also ceased using ARI and SCS for quantitative comparisons. Furthermore, we have visually compared the SPIRAL-alignment and PASTE by displaying the spatial arrangements of SPIRAL-integration derived clusters on the aligned coordinates (Figure S5A in our revised manuscript).

Following your suggestions, we have provided visualizations of the expression pattern of maker genes to validate the annotations. The updated results were shown in the supplementary figures of the cover letter (Figure S1 and S2).

6. The statement "It is striking that the finer structure of RMS, which would not have been found by sole analysis of the low-resolution data (Figure S5A), was revealed by joint clustering (cluster 7 of Figure 6D)." seems not appropriate. The finer structure of RMS has been found by sole analysis with many published spatial domain identification tools, such as DeepST, STAGATE, SpaceFlow.

Reply: We appreciate the reviewer for this question, and we apologized for the previous inappropriate description. First, we would like to clarify that the Stereo-seq data used in our

manuscript has an approximate resolution of 35um, which differs from the STAGATE-utilized data with an approximate resolution of 25um. Upon examining STAGATE's parameters, we found that the omission of the RMS structure was due to its Louvain resolution of 0.8 from STAGATE. In the original STAGATE paper, this setting was specifically designed for the analysis of Stereo-seq data with a resolution of 25um. Increasing the Louvain resolution to 1.2 enabled STAGATE to detect the RMS structure in a single sample. When SPIRAL integrates diverse SRT samples, such as Slide-seq v2 with 10um resolution and Stereo-seq with approximately 35um resolution, it consistently demonstrated superior batch integration performance (Figure 6C in revised manuscript). Notably, SPIRAL outperformed harmony_STAGATE in detecting common structures like RMS, even when the Louvain resolution in harmony_STAGATE was increased to 1.5 (Figure S5B of our revised manuscript).

Based on the analysis above, we have removed the previous inappropriate statement, and clarify the superior performance of SPIRAL in joint clustering on Page 11 of our revised manuscript.

Spatial arrangements of derived clusters by SPIRAL-integration and harmony_STAGATE on Slide-seq v2 and Stereo-seq data with 35um resolution. (6C) displays the common clusters by SPIRAL-integration and (S5B) displays the results of harmony_STAGATE. Each method contains two panels, corresponding to two technologies, and the colors represent clusters.

7. I think only comparison to STAGATE but no other spatial clustering and embedding methods are not valid and sufficient. Even so, I doubt whether the authors well re-implemented it. The performance like Figures 3, 4, 5, and Figure S5C are invalid, and some are not consistent with the original one in STAGATE (<https://stagate.readthedocs.io/en/latest/AT2.html>). The authors may need to double-check if they correctly re-used this method. To make the comparison fair, the authors are supposed to re-run the analysis of harmony_STAGATE and provide the details of running competing methods. Anyway, the benchmarking is still in its stage.

Reply: We thank the reviewer for the comments and concerns regarding the comparison of STAGATE and other methods in our study. Following your suggestion in the first question, we have incorporated further comparisons with spatial clustering and embedding methods, including DeepST, GraphST, STAligner and BASS. In summary, the results consistently demonstrated the superiority of SPIRAL-integration, which have been mentioned in the first question.

Regarding the comment about the implementation of harmony_STAGATE, we have checked it carefully to ensure fairness. We can confirm that we have run the tools correctly. To ensure reproducibility, we have hosted all analysis code used in the present study on https://github.com/guott15/SPIRAL/tree/main/Comparison/STAGATE_cases.ipynb. Additionally, we would like to clarify that

- 1) For the results of DLPFC data in Figure 3, we performed within-donor integrations using STAGATE alone and across-donor integrations with harmony_STAGATE. The original

STAGATE paper presented integration results for donor 3, with a mean ARI score of 0.62, which is consistent with our results. Based on LISI-CoM and ARI results (revised Figure 3B), we concluded that, harmony_STAGATE exhibited relatively high ARI but low LISI-CoM.

3B

Boxplot of LISI-CoM and ARI within and across donors

- For the sagittal mouse brain data, STAGATE did not provide the integration results. Our analysis in Figure 4 revealed that harmony_STAGATE failed to detect the common domains surrounding adjacent regions of the posterior and anterior parts (Figure A below). In contrast, these domains were successfully identified by SPIRAL-integration (red box in revised Figure 4D)

Spatial arrangements of derived clusters by harmony_STAGATE (A) and SPIRAL (4D) on sagittal mouse brain. Each method contains two panels, corresponding to the anterior and posterior parts.

- For the coronal mouse brain in Figure 5, we were also able to reproduce the results of STAGATE in identifying the domains of ‘10X_DAPI’ sample (Figure A below), as depicted in Figure 5 of the original STAGATE paper. However, harmony_STAGATE failed to integrate three samples sequenced by different 10X Visium protocols, including a fresh sample with H&E staining images (‘10X_Normal’), a fresh sample with DAPI staining images (‘10X_DAPI’) and a FFPE sample with H&E staining images. This was shown by the UAMP visualizations (revised Figure S4B & Figure B below) and the delineation of hippocampus region (revised Figure S4D), which indicated that SPIRAL accomplished effective batch effect correction and common domain identification.

Performance comparisons of on coronal mouse brain samples. (A) displays the domains by STAGATE on '10X_DAPI' sample. (S4B&B) displays the UMAP visualizations of SPIRAL (S4B) and harmony_STAGATE (B), different colors represent different protocols, (S4D) spatial arrangements of derived clusters on the region of hippocampus.

4) For Figure S5C, it is important to note that the Stereo-seq data utilized in our manuscript has an approximate resolution of 35um, differing from the STAGATE-utilized data with an approximate resolution of 25um. Therefore, variations in results are anticipated. We displayed the spatial arrangements of Slide-seq v2 and Stereo-seq SRT data with approximately 35um resolution. As previously mentioned in question 6, harmony_STAGATE's performance was weaker than that of SPIRAL-integration. We further integrated Slide-seq v2 and STAGATE-utilized Stereo-seq SRT data (Figure A&B below), and found that both SPIRAL-integration and harmony_STAGATE achieved effective joint domain identifications.

Integration performance of Slide-seq v2 and STAGATE-utilized Stereo-seq SRT data. (A) displays the domains derived from SPIRAL. (B) displays the domains derived from harmony_STAGATE

8. Some analysis and demonstration can be improved. For example, Figure 6A is a bit redundant with two replicated subplots, and too much blank in the third. The plot of Slide-seq V2 in Figure 6D is problematic, maybe there are some noisy parts (not real tissue) therein that should be filtered.

Reply: We thank the reviewer for these detailed comments. We carefully revised Figure 6A accordingly: 1) We replaced the anatomy image with the H&E image from Stereo-seq sample [5]. 2) We rearranged the layout. In addition, we removed the noisy points of Slide-seq V2 sample in SPIRAL, harmony_STAGAGTE and STAligner, as shown in Figure 6C, S5B & S5C of the revised manuscript. The conclusion remains the same.

Minor Comments for the Author

1. It seems to me all the supplementary figures do not have captions.

Reply: We thank the reviewer for this suggestion. In fact, all the captions of supplementary figures can be found in the 'FigS_legend.pdf'.

Reference

- [1] Guo, T., Chen, Y., Shi, M., Li, X., & Zhang, M. Q. (2022). Integration of single cell data by disentangled representation learning. *Nucleic acids research*, 50(2), e8-e8.
- [2] Stuart, T., Butler, A., Hoffman, P., Hafemeister, C., Papalexi, E., Mauck III, W. M., ... & Satija, R. (2019). Comprehensive integration of single-cell data. *Cell*, 177(7), 1888-1902.
- [3] Lanjakornsiripan D, Pior B J, Kawaguchi D, et al. Layer-specific morphological and molecular differences in neocortical astrocytes and their dependence on neuronal layers[J]. *Nature communications*, 2018, 9(1): 1623.
- [4] DeFelipe J, Alonso-Nanclares L, Arellano JJ. Microstructure of the neocortex: comparative aspects. *J Neurocytol*. 2002;31:299–316.
- [5] Chen A, Liao S, Cheng M, et al. Spatiotemporal transcriptomic atlas of mouse organogenesis using DNA nanoball-patterned arrays[J]. *Cell*, 2022, 185(10): 1777-1792. e21.

Cplx3

Cux2

Fezf2

Rorb

A

Cplx3

Cux2

Fezf2

Rorb

B

Pde1c

Calb1

Slc1a2

Gfap

C

D

Pvalb

Pvalb

Slc17a6

Slc17a6

E

F

Plp1

Mbp

Mobp

G

H

I

J

Gene expression visualization of sagittal mouse brain SRT data. **(A&B)** markers of cortex layers on anterior (A) and posterior (B); **(C)** markers of cerebellum on posterior; **(D)** markers of Astrocyte on anterior and posterior; **(E&F)** markers of inhibitory (E) and excitatory (F) neurons. **(G&H)** markers of oligodendrocyte on posterior (G) and anterior (H); **(I&J)** markers of hippocampus (I) and CA (J).

A

B

C

Cplx3

Rprm

Fezf2

Rorb

D

Cplx3

Rprm

Fezf2

Rorb

E

Cplx3

Rprm

Fezf2

Rorb

F

Slc1a2

Slc1a2

Slc1a2

G

Slc17a6

Slc17a6

Sst

Sst

H

I

Gene expression visualization of coronal mouse brain SRT data. **(A, B&C)** markers of hippocampus on '10X_Normal' (A) and '10X_DAPI' (B) and '10X_FFPE' (C); **(C)** markers of cerebellum on posterior; **(D)** markers of Astrocyte on anterior and posterior; **(D,E&F)** markers of cortex layer on '10X_Normal' (D) and '10X_DAPI' (E) and '10X_FFPE' (F); **(G)** markers of Astrocyte; **(H)** markers of excitatory neurons; **(I)** markers of inhibitory neurons. **(J,K&L)** markers of oligodendrocyte on '10X_Normal' (J) and '10X_DAPI' (K) and '10X_FFPE' (L)

Second round of review

Reviewer 1

The authors have revised the manuscript and adequately answered my comments. In its current form, I would recommend it for publication.